# Geometric Contact Flows: Contactomorphisms for Dynamics and Control

**Andrea Testa** [1 2]  **Søren Hauberg** [3]  **Tamim Asfour** [2]  **Leonel Rozo** [1]

## Abstract

Accurately modeling and predicting complex dynamical systems, particularly those involving force exchange and dissipation, is crucial for applications ranging from fluid dynamics to robotics, but presents significant challenges due to the intricate interplay of geometric constraints and energy transfer. This paper introduces Geometric Contact Flows (GFC), a novel framework leveraging Riemannian and Contact geometry as inductive biases to learn such systems. GCF constructs a latent contact Hamiltonian model encoding desirable properties like stability or energy conservation. An ensemble of contactomorphisms then adapts this model to the target dynamics while preserving these properties. This ensemble allows for uncertainty-aware geodesics that attract the system's behavior toward the data support, enabling robust generalization and adaptation to unseen scenarios. Experiments on learning dynamics for physical systems and for controlling robots on interaction tasks demonstrate the effectiveness of our approach.

## 1. Introduction

Modeling dynamical systems is fundamental to many scientific and engineering disciplines, enabling the analysis and prediction of system behaviors across diverse applications (Yu & Wang, 2024). This involves generating state–space trajectories to forecast the system evolution. While the increasing availability of data has accelerated the adoption of data–driven methods, purely black-box models face substantial challenges (Chen et al., 2018). As exemplified in Fig. 1, a naive MLP network fails to capture the true dynamics, leading to inaccurate predictions even within the data

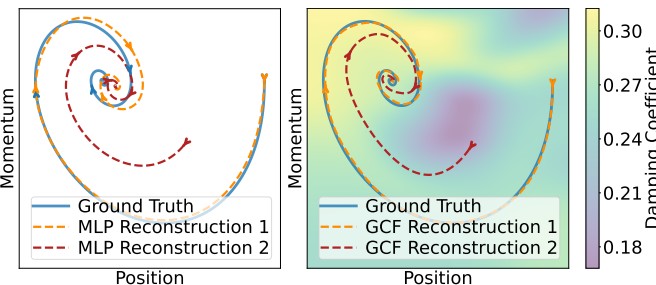

*Figure 1.* A dynamical system modeled via an MLP (left) and our GCF (right). The first prediction (orange) starts within the data support and the second (red) outside. The MLP fails to reconstruct the dynamics and does not converge to the attractor, while GCF provides reliable and physically–interpretable predictions due to its contact Hamiltonian structure.

support (orange trajectory). This stems from the model's inability to encode the physical relationships between variables, which are governed by an underlying energy function. The model also extrapolates unreliably in data–sparse regions (red trajectory), showing the lack of physical interpretability and generalization capacity in black-box approaches (Wang et al., 2020). To overcome these limitations, physics–informed models incorporate physical principles into the model architecture, to ensure consistency with the underlying dynamics. These physical inductive biases appear as geometric structures, such as symplectic and contact manifolds (Blair & Blair, 2010), providing a powerful differential geometric perspective that we exploit.

**This paper** introduces GEOMETRIC CONTACT FLOWS (GFC), a novel framework leveraging Riemannian and contact geometry as inductive biases to learn complex dynamical systems. By integrating these geometries (Sec. 3) with a learning model, GCF provides robust, interpretable, and physically–grounded dynamics modeling, overcoming the limitations of black-box methods. First, a baseline latent dynamics encodes desired properties about the target dynamics (e.g., stability, energy conservation) via a contact structure and an associated Riemannian metric. We then adapt this baseline model to the target dynamics through a contact diffeomorphism (Sec. 4). We improve the GCF generalization robustness via an ensemble of contact diffeomorphisms to identify uncertain regions, allowing the model to adapt its behavior accordingly. This is achieved by modifying the Riemannian metric to incorporate uncertainty, resulting in

[1]Bosch Center for Artificial Intelligence, Renningen, Germany [2]Institute for Anthropomatics and Robotics, Karlsruhe Institute of Technology (KIT), Karlsruhe, Germany [3]Section of Cognitive Systems, Technical University of Denmark (DTU), Lyngby, Denmark. Correspondence to: Andrea Testa <andrea.testa@de.bosch.com>.

*Proceedings of the $42^{nd}$ International Conference on Machine Learning*, Vancouver, Canada. PMLR 267, 2025. Copyright 2025 by the author(s).

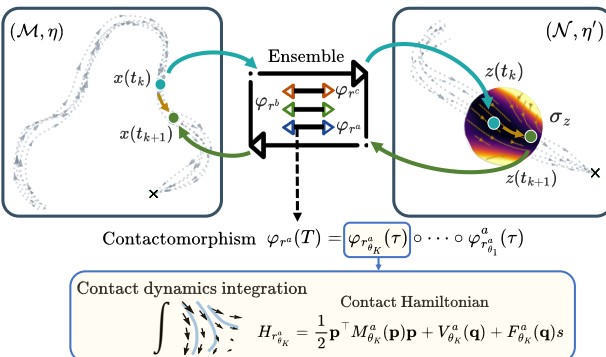

*Figure 2.* Geometric Contact Flows: A state $x(t_k)$ in the ambient space $(\mathcal{M}, \eta)$ is mapped to a corresponding state $z(t_k)$ in the latent space $(\mathcal{N}, \eta')$ via the ensemble of contactomorphisms $\{\varphi_{r^n}(T)\}_{n \in [1, N]}$. The uncertainty $\sigma_z$ estimated by the ensemble is depicted as a colored patch (dark and light colors denote low and high uncertainty). The latent dynamics is integrated by following a contact Hamiltonian model resulting into the latent state $z(t_{k+1})$, which is then projected back into the ambient space as $x(t_{k+1})$ via $\{\varphi_{r^n}^{-1}(T)\}_{n \in [1, N]}$. Each contactomorphism in the ensemble is a composition of transformations (Eq. (10a)), where each transformation results from integrating the flow of a contact Hamiltonian vector field (Eq. (11)).

geodesics that avoid uncertain regions. Furthermore, this Riemannian metric reshaping principle is leveraged to avoid unsafe regions or unexpected interactions (Sec. 5). Figure 2 illustrates the overall GCF architecture. We demonstrate the effectiveness of GCF on two physical dynamical system reconstruction experiments (spring mesh and quantum system), on handwriting dynamics reconstruction using two datasets (Lemme et al., 2015; Fabi et al., 2022), and on two real-world robotic tasks (Sec. 6).

In summary, **our contributions** are: *(1)* Design of latent contact Hamiltonian dynamics with desirable properties that are transferred to the target dynamics; *(2)* An expressive class of neural networks to represent contactomorphisms, preserving the dynamic and geometric properties of the latent dynamics while adapting them to the observed data; *(3)* An ensemble of contactomorphisms to quantify the uncertainty in dynamics prediction; *(4)* A generalization mechanism based on Riemannian geodesics and ensemble uncertainty to guide the dynamical system's behavior and *(5)* Experiments including comparisons to state-of-the-art approaches and a methodology to transform dynamic data to canonical contact Hamiltonian coordinates.

## 2. Related Work

**Diffeomorphisms Learning.** Differential geometry gives a formal framework to connect the target dynamics to a simpler latent space representation through a diffeomorphism (Franks, 1971). A prevalent approach for learning

dynamical systems is to define a simple (stable) dynamical system in a latent space and use a diffeomorphism to map it to the target dynamics, while preserving key latent properties like periodicity or convergence to an attractor (Perrin & Schlehuber-Caissier, 2016; Rana et al., 2020; Zhang et al., 2022). Extensions of this approach incorporate contraction properties to ensure stronger stability guarantees (Mohammadi et al., 2024; Jaffe et al., 2024). However, these methods are limited to first-order dynamical systems and cannot model complex behaviors such as self-crossing trajectories, physical interactions, and asymmetric obstacle avoidance. While Fichera & Billard (2024) extend diffeomorphisms to purely dissipative second-order dynamical systems, their approach remains limited, as it cannot model dynamics with limit cycles or non-zero curl components. We overcome these limitations by introducing a contact Hamiltonian structure in the diffeomorphisms to preserve complex latent system's physical properties, even for general second-order systems.

**Riemannian Metric Learning.** Inspired by the manifold hypothesis (Bengio et al., 2013), latent models can access and represent data manifolds in high-dimensional spaces. This can be exploited to learn a Riemannian metric from dynamic trajectories, subsequently represented as geodesics (Arnold, 1966). This metric also distinguishes the data support from the surrounding ambient space (Hauberg, 2018). Building on this, Beik-Mohammadi et al. (2023) learned a Riemannian metric that takes large values at the boundaries of the data manifold, confining the learned dynamics to a safe, data–informed region. By adjusting the metric, the system's behavior can directly be adapted to specific environments, such as navigating around obstacles (Zhi et al., 2022). Building on this idea, we estimate an uncertainty-based metric to guide the system's geodesics towards safe regions.

**Hamiltonian Learning.** The Hamiltonian framework models second-order dynamics by describing how conjugate variables (e.g., position and momentum) evolve through a symplectic structure, which encodes energy conservation geometrically. While many learning approaches parametrize the energy function to model dynamical systems (Greydanus et al., 2019; Zhong et al., 2020a), or incorporate a symplectic structure into neural networks (Jin et al., 2020; Li et al., 2020; Tong et al., 2021), modeling systems with friction and energy exchange requires extending the Hamiltonian formulation. Several studies (Sosanya & Greydanus, 2022; Zhong et al., 2020b; Chen et al., 2021) incorporate forces and damping as external disturbances on the symplectic manifold. However, these modifications disrupt the symplectic structure and prevent us from conserving the physical properties of the latent system solely through the preservation of the geometric structure via diffeomorphisms. Canizares et al. (2024) maintain the

symplectic structure by doubling the manifold's dimension to represent non-conservative dynamics, at the cost of introducing physically–meaningless variables. We approach this problem by leveraging contact Hamiltonian dynamics.

**Contact Hamiltonian Learning.** Contact Hamiltonian theory (Kholodenko, 2013) extends the symplectic manifold by introducing a single, physically meaningful variable, to account for energy dissipation and generation. This makes contact geometry particularly well-suited for modeling non-conservative systems (Bravetti et al., 2017). The term *contact* refers to the geometric relationship between the dynamical solutions and the energy constraints, where solutions *"touch"* the constraints but do not violate them (Geiges, 2001). Zadra (2023) proposes to parametrize the contact Hamiltonian with a neural network to model a broad class of dynamical systems, but this is unstable beyond the training data, hindering its generalization capabilities. Our approach overcomes these limitations using the contact Hamiltonian framework as an inductive bias to design contact diffeomorphisms. These are later employed to design uncertainty-aware geodesics for robust generalization.

# 3. Preliminaries

**Vector Fields and 1-Forms.** Let $\mathcal{M}$ be a smooth compact manifold. The tangent space at $x \in \mathcal{M}$, denoted $\mathcal{T}_x\mathcal{M}$, consists of all tangent vectors $v$ to $\mathcal{M}$ at $x$. The tangent bundle $\mathcal{T}\mathcal{M}$ is the union of all tangent spaces. An element of $\mathcal{T}\mathcal{M}$ is a pair $(x, v)$. The assignment of a specific tangent vector to every point in the manifold is described by a vector field $X : \mathcal{M} \to \mathcal{T}\mathcal{M}$. Given a function $f : \mathcal{M} \to \mathbb{R}$, the 1-form $df : \mathcal{T}\mathcal{M} \to \mathbb{R}$ generalizes the gradient from Euclidean spaces by measuring the variation of $f$ in $x$ along a direction identified by $v$. The set of all the 1-forms $df \ \forall (x, v)$ is the cotangent bundle $\mathcal{T}^*\mathcal{M}$, the union of all the cotangent spaces $\mathcal{T}_x^*\mathcal{M}$. The contact and Riemannian geometries provide two distinct mechanisms to associate a 1-form to a vector field, thereby establishing different connections between the tangent and cotangent bundles. In simpler terms, these geometries link the same function $f$ to different vector fields $X$, as illustrated in Fig. 3.

**Riemannian Geometry.** A Riemannian metric $g : \mathcal{T}\mathcal{M} \times \mathcal{T}\mathcal{M} \to \mathbb{R}$ is a smooth, symmetric, and positive-definite bilinear map that defines an inner product on the tangent spaces. The length of a smooth curve $x(t) : [t_0, t_1] \to \mathcal{M}$ w.r.t. the metric $g$ is $l = \int_{t_0}^{t_1} \sqrt{g(\dot{x}(t), \dot{x}(t))}dt$, where $\dot{x}(t) \in \mathcal{T}_{x(t)}\mathcal{M}$ is the vector tangent to the curve at $x(t)$. The curve minimizing this length between two points $x(t_0)$ and $x(t_1)$ on $\mathcal{M}$ is called a *geodesic*. Geodesics generalize straight lines in Euclidean space to curved spaces, representing the shortest paths in the geometry induced by $g$. The geometric structure $g(\dot{x}(t), \dot{x}(t))$ does not need to be symmetric with respect to $\dot{x}$ to measure curve lengths. In

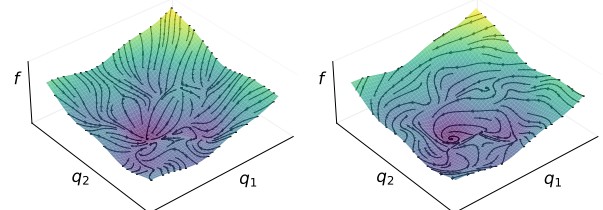

*Figure 3.* The same scalar function $f$, via its 1-form $df$, yields two distinct vector fields (shown as streamlines) under different geometric structures. In Riemannian geometry (left), the streamlines correspond to the gradient flow, following the direction of steepest ascent or descent of $f$, acting as a potential. In contrast, in contact geometry (right), the function $f$ describes an energy, and the angle between the streamlines and the level curves of $f$, depends on the energy dissipation in the dynamics, governed by Eq. (8).

such cases, it defines a Finslerian structure, generalizing the Riemannian structure (Chern, 1996). The Riemannian metric $g$ establishes a correspondence between a differential 1-form $df$ and a vector field $X_f$ through the relation,

$$df(X) = g(X_f, X), \quad \forall X \in \mathcal{T}\mathcal{M}. \tag{1}$$

The vector field $X_f$ is orthogonal to the level sets of $f$, which can be interpreted as a potential function (Fig. 3).

**Contact Geometry.** A 1-form $\eta$ on a $(2d+1)$-dimensional manifold $\mathcal{M}$ is called a contact form if it satisfies the maximal non-integrability condition (Geiges, 2008). This means that the wedge product of $\eta$ with the $d$-times repeated wedge product of its exterior derivative $d\eta$, is non-zero everywhere on $\mathcal{M}$: $\eta \wedge (d\eta)^d \neq 0$. A *contact manifold* is then the pair $(\mathcal{M}, \eta)$. Given the 1-form $df$, the contact form $\eta$ uniquely determines a vector field $X_f$ through the relation,

$$df = d\eta(X_f, X) - \mathcal{L}_{X_f}\eta(X), \ \forall \ X \text{ in } \mathcal{T}\mathcal{M}, \tag{2}$$

where $\mathcal{L}_{X_f}$ denotes the Lie derivative along $X_f$ (Geiges, 2001). The non-integrability condition ensures that $\eta$ imposes nonholonomic constraints, restricting the allowable directions of motion without confining the dynamics to a submanifold. Consequently, unlike the symplectic case, motion is not constrained to the level sets of $f$ (Cruz, 2018). In both symplectic and contact geometry, $f$ is typically referred to as the Hamiltonian $H$ and interpreted as energy, with the associated vector field written as $X_H$. Consequently, while symplectic geometry is limited to describing conservative dynamical systems, contact geometry offers a more general framework for modeling a broader range of dynamical behaviors (Bravetti et al., 2017).

The integral flow $x(t)$, generated by the vector field $X_H$ from an initial point $x_0$, defines a one-parameter family of transformations $\varphi(t)$, mapping $x_0 \to x(t)$ within the contact manifold $(\mathcal{M}, \eta)$:

$$x(t) = \varphi(t)(x_0) : [0, T] \subset \mathbb{R} \to (\mathcal{M}, \eta). \tag{3}$$

Each transformation $\varphi(t) : (\mathcal{M}, \eta) \to (\mathcal{M}, \eta')$ is smooth, invertible, and preserves the contact structure up to a conformal factor, i.e., $\varphi^* \eta = a\eta'$, $a \in \mathbb{R}$ (Zadra, 2023). So, $\varphi(t)$ constitutes a one-parameter family of contact-preserving diffeomorphisms, a.k.a. *contactomorphisms*. More generally, a contactomorphism maps between distinct contact manifolds $\varphi : (\mathcal{M}, \eta) \to (\mathcal{M}, \eta')$. In GCFs, contactomorphisms are employed both to represent integral dynamic contact flows $x(t)$ within a single manifold and as a mapping between ambient and latent spaces.

**Contact Flows as Riemannian Geodesics.** Notably, the dynamic contact flow $x(t)$ on the contact manifold $(\mathcal{M}, \eta)$ can be interpreted as a reparametrization of a geodesic on an augmented space–time Riemannian manifold $(\mathcal{R} \times \mathbb{R}, g)$, where $\mathcal{M} = \mathcal{T}^* \mathcal{R} \times \mathbb{R}$ (Abraham & Marsden, 2008). In this setting, we denote the state variables as $q \in \mathcal{R}$, $\dot{q} \in \mathcal{T}_q \mathcal{R}$, $p \in \mathcal{T}_q^* \mathcal{R}$, and $s \in \mathbb{R}$, with $x = \{q, p, s\}$. This means that the augmented Riemannian manifold is equipped with a cotangent bundle $\mathcal{T}^* \mathcal{R}$ endowed with a contact structure. This connection between the contact and the Riemannian geometries arises from Maupertuis' principle (López & Martínez, 2000), which first adopts a variational perspective by expressing the contact flow $x(t)$ as the solution of a dynamic optimization problem,

$$\min_{q,p,s} \int_{t_0}^{t_1} p(t) \cdot \dot{q}(t) - H(q(t), p(t), s(t)) \, dt. \quad (4)$$

This formulation, defined on $\mathcal{T}^* \mathcal{R} \times \mathbb{R}$, can be transferred to $\mathcal{T} \mathcal{R} \times \mathbb{R}$ by mapping the 1-form $p$ to vector $\dot{q}$ via the Riemannian connection in Eq. (1), resulting in $\min_{\dot{q}} \int_{t_0}^{t_1} \mathcal{L}(q(t), \dot{q}(t)) \, dt$, where $\mathcal{L}$ is the transformed cost functional, i.e., the Lagrangian. This dynamic optimization can be reparametrized using $s$ as the independent variable, with $ds = \sqrt{H(q, p, s) - H_V(q)} \, dt$, where $H_V$ denotes the potential energy component of the Hamiltonian (Abraham & Marsden, 2008; Udrişte, 2000). In this setting, $s$ plays the role of arc-length to be minimized on the Riemannian manifold and corresponds to the action associated with the Lagrangian $\mathcal{L}$. The problem then reduces to a geodesic computation on $\mathcal{R} \times \mathbb{R}$,

$$\min_{\dot{q}} \int_{s_0}^{s_1} \sqrt{\hat{g}(\dot{q}(s), \dot{q}(s), s)} \, ds. \quad (5)$$

The modified Riemannian metric $\hat{g}$ is induced by the Hamiltonian $H$ and depends on the variable $s$. This dependence reflects how the shape of trajectories is affected by their own length, capturing the influence of non-conservative forces on the system's evolution. By modifying the metric $\hat{g}$, one can alter the system's dynamics, providing a geometric approach for controlling or shaping its trajectories (Udrişte, 2003; López & Martínez, 2000). The resulting geodesics lie on $\mathcal{R} \times \mathbb{R}$, as they depend on the reference level of $s$. They can be projected onto the configuration space $\mathcal{R}$ by

fixing this value (Udrişte, 2000). In symplectic geometry for conservative systems, $s$ remains constant and is not a dynamic variable. In contrast, contact geometry captures energy dissipation or external input through the evolution of $s$. Our proposed Geometric Contact Flows build on this dual perspective, modeling the dynamics by harnessing the contact interpretation while generalizing it through the Riemannian perspective. Further details on these concepts, along with a broader comparison between contact and symplectic structures, are provided in App. A.

# 4. Contactomorphisms Learning

**Latent Contact Hamiltonian Dynamics.** Let $(\mathcal{N}, \eta')$ be a $(2d + 1)$-dimensional contact manifold, locally defined by a set of canonical coordinates $\mathbf{z} = \{\mathbf{q}, \mathbf{p}, s\}$, with $\mathbf{q} = \{q_1, \ldots q_d\}$, $\mathbf{p} = \{p_1, \ldots p_d\}$. In classical mechanics, the conjugate variables $q_i$ and $p_i$ respectively represent position and momentum, while $s$ is the Lagrangian action. As detailed in Sec. 3, the contact manifold can be interpreted as the augmented cotangent bundle of a Riemannian manifold $\mathcal{R}_{\mathcal{N}}$, i.e., $\mathcal{N} := \mathcal{T}^* \mathcal{R}_{\mathcal{N}} \times \mathbb{R}$, with $\mathbf{q} \in \mathcal{R}_{\mathcal{N}}$, $\{\mathbf{q}, \mathbf{p}\} \in \mathcal{T}^* \mathcal{R}_{\mathcal{N}}$ and $s \in \mathbb{R}$. A contact Hamiltonian dynamics on the latent space $(\mathcal{N}, \eta')$ is defined via a contact Hamiltonian function $H_g(\mathbf{z})$, generating a flow $\varphi_g$ such that,

$$\mathbf{z}(t) = \varphi_g(t)(\mathbf{z}(0)), \quad t \in [0, t_1]. \quad (6)$$

The choice of $H_g(\mathbf{z})$ induces a metric $\hat{g}$ on $\mathcal{R}_{\mathcal{N}}$, making $\varphi_g$ follow its geodesics.

We propose to preserve the properties of the dynamical system $H_g$ in the ambient space through a suitable coordinates change, i.e., via a contactomorphism. The choice of the latent dynamics does not affect the reconstruction of the target dynamics on the data support, but instead serves as an inductive bias to guide the generalization. Possible choices for $H_g$ are,

$$H_{g^A} = \frac{1}{2} \mathbf{p}^\top \mathbf{p} + \frac{1}{2} \mathbf{q}^\top \mathbf{q}, \quad (7a)$$

$$H_{g^B} = \frac{1}{2} \mathbf{p}^\top \mathbf{p} + \frac{1}{2} \mathbf{q}^\top \mathbf{q} + s, \quad (7b)$$

$$H_{g^C} = \left( \frac{1}{2} \mathbf{p}^\top \mathbf{p} + \frac{1}{2} \mathbf{q}^\top \mathbf{q} + s \right) s^2. \quad (7c)$$

For instance, to ensure periodic orbits in the target dynamics, we can use the simple harmonic oscillator (7a), as our latent dynamics. Other dynamical systems, such as the example shown in Fig. 1, naturally converge to an attractor. This behavior can be induced in the latent dynamics by incorporating a damping term. We use the dependence of the contact Hamiltonian on the Lagrangian action $s$, to govern the variation of the system's total energy. This relationship is expressed as follows (Bravetti et al., 2017),

$$H(t) = H(0) \, e^{\int_0^t \partial H / \partial s \, d\tau}, \quad (8)$$

where $\partial H / \partial s$ represents the damping coefficient of the dynamics. Consequently, the $+s$ term in Eq. 7b ensures the depletion of the system energy and the convergence toward the attractor $\{\mathbf{q}, \mathbf{p}, s\} = \mathbf{0}$. Instead, the dynamics (7c) is suitable for safety-critical scenarios requiring stopping the system near unsafe regions of the state space $(\mathcal{N}, \eta')$. By scaling all terms of the contact Hamiltonian with $s$, the energy $H$ can reach zero while $\mathbf{q}$ and $\mathbf{p}$ remain non-zero, stopping the system before it reaches the attractor. Unsafe regions are here characterized by low values of the Lagrangian action $s$.

**Contactomorphisms Design.** While diffeomorphisms are commonly used to align latent and target dynamics in first-order systems, they fail to preserve the intrinsic structure of second-order systems, in particular the interplay between conjugate variables encoded in the contact form. To address this, we leverage contactomorphisms, which preserve $\eta'$ and thus maintain the latent space structure. The practical implications of this inductive bias are examined in the ablation study presented in App. C.1. A contactomorphism $\varphi$ transforms coordinates $\mathbf{z}$ from the latent space $(\mathcal{N}, \eta')$ into a new set of coordinates $\mathbf{x}$ that describe the ambient space $(\mathcal{M}, \eta)$. As described in Sec. 3, we model this transformation as the flow $\varphi_r(t)$ of a contact Hamiltonian system evaluated at time $t = T$,

$$\mathbf{z} = \varphi_r(T)(\mathbf{x}), \quad \mathbf{x} = \varphi_r^{-1}(T)(\mathbf{z}). \quad (9)$$

For expressivity, we implement this contact transformation as a composition of $K$ sequential parametrized networks,

$$\varphi_r(T) = \varphi_{r_{\theta_K}}(\tau) \circ \cdots \circ \varphi_{r_{\theta_k}}(\tau) \circ \cdots \circ \varphi_{r_{\theta_1}}(\tau), \quad (10a)$$

$$\varphi_r^{-1}(T) = \varphi_{r_{\theta_1}}^{-1}(\tau) \circ \cdots \circ \varphi_{r_{\theta_k}}^{-1}(\tau) \circ \cdots \circ \varphi_{r_{\theta_K}}^{-1}(\tau). \quad (10b)$$

Each individual contactomorphism $\varphi_{r_{\theta_k}}(\tau)$ updates the initial point $\mathbf{x}$ by integrating, for a duration $\tau$, the vector field associated to the contact Hamiltonian $H_{r_{\theta_k}}$, defined as,

$$H_{r_{\theta_k}} = \frac{1}{2}\mathbf{p}^\top M_{\theta_k}(\mathbf{p})\mathbf{p} + V_{\theta_k}(\mathbf{q}) + F_{\theta_k}(\mathbf{q})s, \quad (11)$$

where $M_{\theta_k}(\mathbf{p}), V_{\theta_k}(\mathbf{q}), F_{\theta_k}(\mathbf{q})$ are scalar functions of the conjugate variables, parametrized by random Fourier features networks (Rahimi & Recht, 2007). The structure of $H_{r_{\theta_k}}$ and the choice of network architecture are elaborated in the ablation studies of App. C.3. The integration of the associated vector field is performed using a contact splitting integrator (Zadra, 2023), as detailed in App. B.2. Our architecture is analytically invertible, minimizing the computational cost of computing $\varphi_{r_{\theta_k}}^{-1}(\tau)$.

Training a contactomorphism $\varphi_r(T)$, given a dataset of $B$ trajectories from the target dynamics $\{\bar{\mathbf{x}}_b(t), \ t \in [0, t_b]\}_{b \in [1,B]}$, involves the following steps: (1) We map the training trajectories to their corresponding latent trajectories $\{\bar{\mathbf{z}}_b(t), \ t \in [0, t_b]\}_{b \in [1,B]}$ via the contactomorphism $\varphi_r(T)$; (2) Starting from the initial latent states

$\{\bar{\mathbf{z}}_b(0)\}_{b \in [1,B]}$, we integrate the latent dynamics to obtain the predicted latent trajectories $\{\mathbf{z}_b(t), \ t \in [0, t_b]\}_{b \in [1,B]}$; (3) We then map the predicted latent trajectories to the ambient space using the inverse contactomorphism $\varphi_r^{-1}(T)$, yielding $\{\mathbf{x}_b(t), \ t \in [0, t_b]\}_{b \in [1,B]}$; (4) Finally, we compute the estimation error in both spaces using the average loss,

$$\ell = \frac{1}{B}\sum_{b=1}^{B} \frac{1}{t_b}\sum_{t=0}^{t_b} \Big( w_x \big\|\bar{\mathbf{x}}_b(t) - \mathbf{x}_b(t)\big\|_2^2 + w_z \big\|\bar{\mathbf{z}}_b(t) - \mathbf{z}_b(t)\big\|_2^2 \Big), \quad (12)$$

where $w_x$ and $w_z$ are weights balancing the ambient and latent space components. Typically, $w_x \gg w_z$, as we focus on capturing the ambient dynamics. However, the latent error term acts as a useful regularizer, as shown in App. C.2. Algorithm 1 in App. B.1 outlines the training phase.

After training, predicting the ambient dynamics from an initial point $\mathbf{x}_0$ involves mapping it to the latent space via the learned contactomorphism $\varphi_r$, integrating the latent dynamics $\varphi_g$, and then mapping the result back to the ambient space via the inverse contactomorphism $\varphi_r^{-1}$. Importantly, integrating the latent dynamics does not require evaluating the learned contactomorphism at every step. This enables efficient long-horizon predictions using only a single forward and inverse mapping. Formally, this process is expressed as a composition of contact Hamiltonian flows,

$$\mathbf{x}(t) = \varphi_r^{-1}(T) \circ \varphi_g(t) \circ \varphi_r(T)(\mathbf{x}_0), \ t \in [0, t_1]. \quad (13)$$

Details on the physical interpretation of this composition are provided in App. B.4.

## 5. Contactomorphisms Generalization

**Ensemble of contactomorphisms.** Relying on a single contactomorphism to map the latent dynamics to the ambient space neglects predictive uncertainty. While predictions may closely align with the training trajectories on the data manifold, extrapolation in data-sparse regions tends to be unreliable. To quantify uncertainty and identify the boundaries of the data manifold, we propose to use an ensemble of $N$ contactomorphisms $\{\varphi_{r^n}(T)\}_{n \in [1,N]}$, each randomly initialized and trained on the same dataset (Syrota et al., 2024). In data-rich regions, the predictions from the ensemble closely match, while in regions with limited data, the predictions diverge, resulting in high model uncertainty.

The ensemble-based uncertainty quantification, whether in ambient or latent space, is computed by transforming the starting point of the trajectories using the ensemble of contactomorphisms to obtain $N$ points:

$$\{\mathbf{z}\}_{n \in [1,N]} = \{\varphi_{r^n}(T)\}_{n \in [1,N]}(\mathbf{x}), \quad (14a)$$

$$\{\mathbf{x}\}_{n \in [1,N]} = \{\varphi_{r^n}^{-1}(T)\}_{n \in [1,N]}(\mathbf{z}). \quad (14b)$$

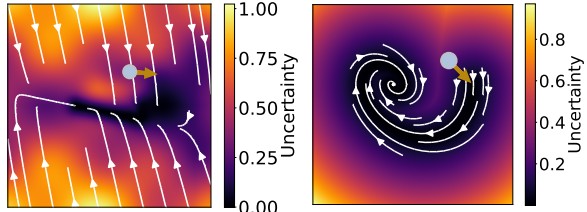

*Figure 4.* Optimal control problem (18) in both latent (left) and ambient spaces (right). The learned dynamics $\dot{\mathbf{z}} = Z_{H_g}$ is extended by the control input $\mathbf{u}$ (represented by the golden arrow) to minimize uncertainty and quickly converge to the data support.

The average prediction is computed and mapped back to the original space to estimate the predictive variance of the resulting $N$ points,

$$\sigma_{\mathbf{x}} = \sigma\{\varphi_{r^n}^{-1}(T)\}_{n\in[1,N]}\left(\mu\{\mathbf{z}\}_{n\in[1,N]}\right), \quad (15a)$$

$$\sigma_{\mathbf{z}} = \sigma\{\varphi_{r^n}(T)\}_{n\in[1,N]}\left(\mu\{\mathbf{x}\}_{n\in[1,N]}\right), \quad (15b)$$

where $\mu$ and $\sigma$ denote the mean and variance of the predictions. Note that this computation requires all contactomorphisms to share the same latent space. We ensure this by jointly training the ensemble as follows: (1) Transform all $B$ training trajectories into $N \times B$ latent trajectories $\{\bar{\mathbf{z}}_b^n(t), \, t \in [0, t_b]\}_{b\in[1,B]}^{n\in[1,N]}$ using all $N$ contactomorphisms $\varphi_{r^n}$; (2) Integrate the latent dynamics starting from the $N \times B$ initial latent states $\{\bar{\mathbf{z}}_b^n(0)\}_{b\in[1,B]}^{n\in[1,N]}$ to compute the predicted latent trajectories $\{\mathbf{z}_b^n(t), \, t \in [0, t_b]\}_{b\in[1,B]}^{n\in[1,N]}$; (3) Randomly select $N$ inverse maps with replacement from the ensemble to compute the ambient predictions $\{\mathbf{x}_b^n(t), \, t \in [0, t_b]\}_{b\in[1,B]}^{n\in[1,N]}$; (4) Compute the average loss over the $N$ contactomorphisms as $\ell_e = \frac{1}{N}\sum_{n=1}^{N}\ell$, with $\ell$ defined as in Eq. 12. Algorithm 2 in App. B.1 summarizes this process.

**Designing uncertainty-aware geodesics.** We leverage the ensemble uncertainty to drive the system dynamics to remain within or converge toward well-informed, data-rich regions. As discussed in Sec. 3, the contact Hamiltonian dynamics $\mathbf{z}(t)$ on the latent space $(\mathcal{N}, \eta')$, governed by the Hamiltonian function $H_g$, can be interpreted as geodesics $\mathbf{q}(s)$ on the augmented Riemannian manifold $(\mathcal{R}_{\mathcal{N}} \times \mathbb{R}, \hat{g})$ with respect to the metric $\hat{g}$ in Eq. (5). To guide the dynamics away from uncertain regions, we propose to reshape $\hat{g}$ using the ensemble uncertainty $\sigma_{\mathbf{z}}$. This increases the traversal cost outside the data manifold, thereby steering the latent dynamics away from data-poor regions. The geodesic $\mathbf{q}(s)$ is obtained by solving the optimization problem,

$$\min_{\dot{\mathbf{q}}} \int_{s_0}^{s_1} \left(\sqrt{\hat{g}(\dot{\mathbf{q}}, \dot{\mathbf{q}}, s)} + \sigma_{\mathbf{z}}(\mathbf{q}, \dot{\mathbf{q}}, s)\right) ds, \quad (16)$$

where canonical coordinates $\mathbf{z} = \{\mathbf{q}, \mathbf{p}, s\} \in \mathcal{T}^*\mathcal{R}_{\mathcal{N}} \times \mathbb{R}$ are converted to $\{\mathbf{q}, \dot{\mathbf{q}}, s\} \in \mathcal{T}\mathcal{R}_{\mathcal{N}} \times \mathbb{R}$. This corresponds to computing a geodesic with respect to an augmented metric

$\hat{g}'$ which is generally Finslerian (see Sec. 3), due to its potential asymmetry in $\dot{\mathbf{q}}$. To solve this, we bring back the optimization to the contact manifold $\mathcal{M} = \mathcal{T}^*\mathcal{R}_{\mathcal{N}}$ by adopting the geodesic interpretation of dynamical systems (Udriste & Udriste, 2000). This involves the reformulation of the problem as a dynamic optimization under a time reparametrization (López & Martínez, 2000). Specifically, we define a perturbation $\mathbf{u}$ to the velocity field of the learned geodesic $\dot{\mathbf{q}}_g$ under metric $\hat{g}$, resulting in a perturbed direction $\dot{\mathbf{q}}_{g'}$ under a modified metric $\hat{g}'$,

$$\min_{\mathbf{u}} \int_{t_0}^{t_1} \left(\sigma_{\mathbf{z}}(\mathbf{q}_{g'}, \dot{\mathbf{q}}_{g'}, s) + \|\mathbf{u}(t)\|^2\right) dt, \quad (17)$$
$$\text{s.t.} \quad \dot{\mathbf{q}}_{g'} = \dot{\mathbf{q}}_g + \mathbf{u}(t).$$

The control term $\mathbf{u}$ locally deforms the trajectory to avoid uncertain regions. Since the learned geodesic flow is encoded by the contact Hamiltonian vector field $Z_{H_g}$, we lift the optimization to the latent state space via the Riemannian connection in Eq. (1),

$$\min_{\mathbf{u}} \int_{t_0}^{t_1} \left(\sigma_{\mathbf{z}}^2(\mathbf{z}(t)) + \|\mathbf{u}(t)\|^2\right) dt \quad (18)$$
$$\text{s.t.} \quad \dot{\mathbf{z}}(t) = Z_{H_g}(\mathbf{z}(t)) + \mathbf{u}(t).$$

In regions where the uncertainty is low (i.e., $\sigma_{\mathbf{z}} \approx 0$), the control action vanishes (i.e., $\mathbf{u} \approx 0$), and the resulting latent dynamics $\dot{\mathbf{z}}(t) = Z_{H_g}(\mathbf{z})$ are unchanged, corresponding to the original latent metric, i.e., $\hat{g}' \approx \hat{g}$. Conversely, in high-uncertainty regions, the control action $\mathbf{u}$ modifies the dynamics by bending the latent trajectories to guide them away from data-poor regions. Figure 4 illustrates the optimal control problem (18) in the reconstruction of the damped harmonic oscillator dynamics (Figure 1).

**Designing safety-aware geodesics.** The optimization problem (18) can be extended with other types of energy functions aimed at penalizing the crossing of unsafe regions. To illustrate this, consider learning a contact Hamiltonian dynamics for a controlled physical system. Here, an unsafe region can be described by obstacles in the system workspace. Formally, an obstacle in the position space $\mathcal{R}_{\mathcal{N}}$ is represented by a set $\Upsilon_{\mathbf{q}}$ consisting of positions $\mathbf{q}_o \in \Upsilon_{\mathbf{q}}$, uniformly sampled from the spatial distribution of points occupied by the obstacle. This set extends to the contact ambient space as $\Upsilon_{\mathbf{x}}$, where each state $\mathbf{x}_o = \{\mathbf{q}_o, \mathbf{p}, s\} \in \Upsilon_{\mathbf{x}} \subset (\mathcal{M}, \eta)$ is obtained by augmenting $\mathbf{q}_o$ with the momentum $\mathbf{p}$ and action variable $s$ that the dynamical system can acquire. This formulation implies that an obstacle at $\mathbf{q}_o$ cannot be traversed for any combination of $\mathbf{p}$ and $s$. The sampled points $\mathbf{x}_o$ are mapped to the latent space $(\mathcal{N}, \eta')$ via the ensemble, as $\Upsilon_{\mathbf{z}} = \{\varphi_{r^n}(T)(\mathbf{x}_o)\}_{n\in[1,N]}$. In this latent space, the energy assigned to each bin is determined by the density of mapped points it contains. This energy distribution is then

incorporated into the optimization (18) as a penalty term to enforce obstacle avoidance,

$$\min_{\mathbf{u}(\cdot)} \int_{t_0}^{t_1} \left( E_\Upsilon(\mathbf{z}(t)) + \sigma_\mathbf{z}^2(\mathbf{z}(t)) + \|\mathbf{u}(t)\|^2 \right) dt \tag{19}$$
$$\text{s.t.} \quad \dot{\mathbf{z}}(t) = Z_{H_g}(\mathbf{z}(t)) + \mathbf{u}(t),$$

where $E_\Upsilon(\mathbf{z}(t))$ denotes the energy term associated with the unsafe region, i.e., the latent space obstacle distribution. This ensures that unsafe latent space regions are associated with high energy values, which penalize trajectories that cross these regions, resulting in e.g., obstacle avoidance.

# 6. Results

We test our approach on established baselines that incorporate inductive biases in learning dynamical systems. Euclideanizing Flows (EF) (Rana et al., 2020) use diffeomorphisms to encode desirable properties like periodicity or target convergence. Neural Contractive Dynamical Systems (NCDS) (Mohammadi et al., 2024) improve upon EF by enforcing global contraction. Hamiltonian Neural Networks (HNN) (Greydanus et al., 2019) embed physical structure by modeling systems with Hamiltonian dynamics. Dissipative Hamiltonian Neural Networks (DHNN) (Sosanya & Greydanus, 2022) extend HNNs to non-conservative systems by introducing dissipation. Details on the experimental setup are provided in App. D. Scalability across target dimensions and model sizes is evaluated in App. E.5. In classical mechanics, the state of physical systems is typically described by positions and velocities $\{q, \dot{q}\}$, which we convert to canonical coordinates $\{q, p, s\}$, required in the contact Hamiltonian formulation (App. D.3). The Lagrangian action $s$ is not directly observable but it is estimated by comparing the system's behavior as described by Maupertuis' principle with that derived from Noether's theorem, ensuring consistency between the two representations.

**Spring Mesh Dynamics Reconstruction.** We consider a 60-dimensional dataset (Otness et al., 2021) describing the dynamics of a 2D square grid of nodes connected by springs. The interaction among multiple springs results in complex, large-scale deformations and oscillations. Predicting the motion of mesh nodes closely resembles finite element modeling of material deformation. Table 1 summarizes the dynamics reconstruction results, reporting the Dynamic Time Warping Distance (DTWD) (Berndt & Clifford, 1994) across 20 trials on systems simulated for 8 seconds from varying initial conditions. Our GCF model achieves a 57% reduction in reconstruction error. Experimental settings and predicted dynamics plots are in given in App. E.1.

*Table 1.* Reconstruction error via DTWD on the spring mesh

| GCF | DHNN | HNN | EF | NCDS | MLP |
|---|---|---|---|---|---|
| **0.50**$_{\pm.19}$ | 1.24$_{\pm.62}$ | 1.49$_{\pm.74}$ | 30.9$_{\pm3.3}$ | 24.9$_{\pm2.6}$ | 1.71$_{\pm.56}$ |

**Quantum Dynamics Reconstruction.** A single-mode bosonic system is a quantum mechanical oscillator whose deterministic dynamics are governed by a contact Hamiltonian operator. The evolution of its wave function is described by the quantum operators $\hat{q}$ and $\hat{p}$, representing the system's stochastic position and momentum, along with the quantum phase $s$, which governs interference between wave components. We conducted experiments on 20 synthetic systems with varying parameters, reconstructing the dynamics of the state variables' expected values using the GCF model and baseline methods, over an 8 seconds horizon. HNN and DHNN are inapplicable here due to their limitation to even-dimensional phase spaces, which prevents modeling the additional odd variable $s$. Results are summarized in Table 2, showing that GCF achieves a 60% reduction in reconstruction error. Further details are provided in App. E.2.

*Table 2.* Reconstruction error via DTWD on the quantum system

| GCF | DHNN | HNN | EF | NCDS | MLP |
|---|---|---|---|---|---|
| **0.29**$_{\pm0.04}$ | N/A | N/A | 0.72$_{\pm0.12}$ | 0.70$_{\pm0.11}$ | 0.41$_{\pm0.06}$ |

**Handwriting Datasets.** The LASA (Lemme et al., 2015) and DigiLeTs (Fabi et al., 2022) datasets are widely used benchmarks for motion generation, featuring 2D handwritten trajectories. LASA involves simpler, first-order dynamics, while DigiLeTs includes more complex, second-order motions with self-crossing paths. The learned dynamics are treated as a skill that an actuated system must not only imitate but also generalize and adapt to new scenarios. In these tasks, the dynamics converge to a target, so we exclude HNN, which models only periodic behaviors. Figure 5 compares the remanent approaches on two characters. EF and NCDS model a position-dependent vector field, enabling flow lines to be plotted in the background. In contrast, DHNN includes position and momentum, and GCF incorporates the Lagrangian action, making global flow lines infeasible. Instead, circular patches depict local dynamics around trajectory points with fixed extra-variable values. GCF's patches represent the model uncertainty.

*Table 3.* Reconstruction error via DTWD on handwriting datasets

| Character | EF | NCDS | DHNN | GCF |
|---|---|---|---|---|
| $\curvearrowright$ | **0.43**$_{\pm0.10}$ | 0.44$_{\pm0.14}$ | 0.65$_{\pm0.15}$ | 0.44$_{\pm0.12}$ |
| $\ell$ | 2.22$_{\pm0.03}$ | 2.79$_{\pm0.09}$ | 0.82$_{\pm0.22}$ | **0.70**$_{\pm0.36}$ |

EF and NCDS are limited to first-order dynamics and cannot capture self-crossing trajectories. DHNN lacks convergence guarantees, often failing to reach the target and to generalize beyond the training data. In contrast, GCF accurately reconstructs the dynamics, converges to the data manifold, and reduces uncertainty, as shown by streamlines avoiding high-uncertainty areas. Table 3 reports the reconstruction error of the position dynamics. GCF achieves performance comparable to EF on LASA, while significantly outperforming

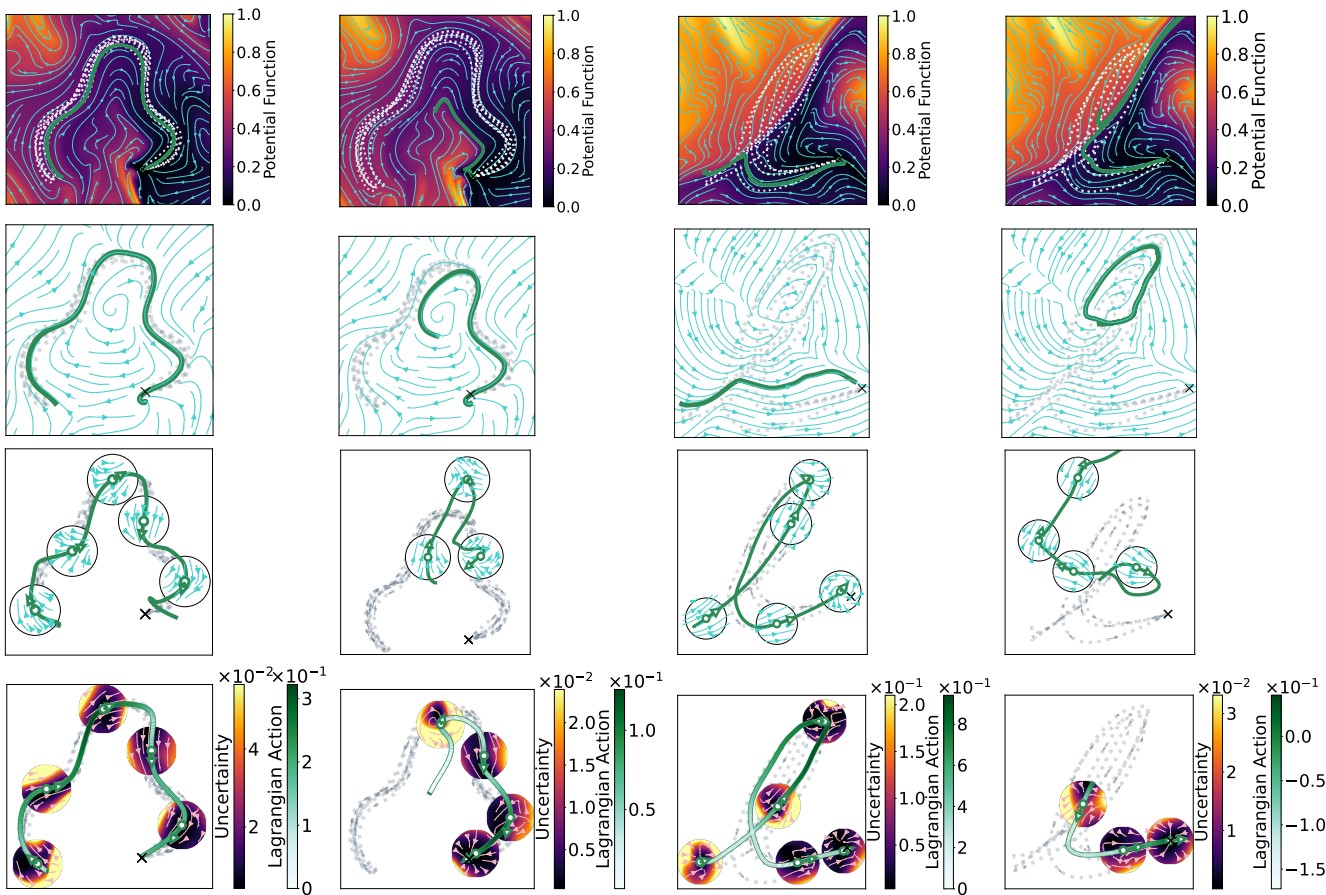

*Figure 5.* Results on the handwriting datasets: Reconstruction and generalization of the LEAF_2 character from the LASA dataset and the ELLE character from the DigiLeTs dataset. From top to bottom rows, EF, NCDS, DHNN and GCF methods. Demonstrations and model predictions are shown as dashed and solid green lines.

other methods on DigiLeTs. Comprehensive results with additional characters are provided in App. E.3. Figure 6 shows generalization tests on two characters, with predictions initialized from states outside the training distribution. A grid of points in the position space is used, with other state variables set to zero. The violin plots provide a statistical analysis of these predictions. Each point, representing a predicted trajectory, is computed as the ratio between time steps spent within the data manifold (in the position space) and the total time steps. Due to the lack of constraints outside the data manifold, only few DHNN trajectories converge, as

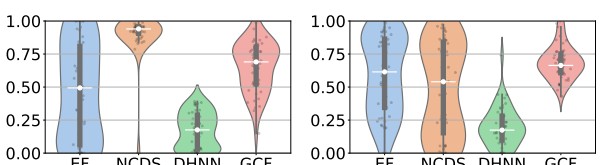

*Figure 6.* Generalization results for the LEAF_2 (left) and ELLE (right) characters. The violin plots depict the distribution of convergence rates for predictions initialized from a grid of points in the state space. While NCDS performs well on LEAF_2, it struggles with ELLE. In contrast, GCF shows greater reliability, achieving the highest average convergence ratio and lowest variance.

shown by the high proportion of points with a ratio of 0. EF ensures asymptotic stability, guiding trajectories to the target but not toward the data manifold, resulting in uniform ratio distributions. NCDS performs better with contractive behavior, but neither method supports second-order dynamics. In contrast, GCF shows greater reliability, ensuring all the dynamics predictions for the two characters converge to the data manifold and spend most of their time within it. Table 4 reports quantitative metrics for the two characters. Figure 7 illustrates how GCF adapts to unseen scenarios. Obstacles, absent during training and representing unsafe regions, are depicted as red points in the position space. Obstacle-induced energy steers the system dynamics away from collisions, as shown by higher cost and deflected streamlines near the obstacle. The right plot compares distances to the data support and the obstacle. The former distance peaks near the obstacle, then drops, marking exit and reentry into

*Table 4.* Generalization measured as average ratio of convergence

| Character | EF | NCDS | DHNN | GCF |
|---|---|---|---|---|
| ⌒ | $0.49_{\pm 0.37}$ | $0.94_{\pm 0.19}$ | $0.18_{\pm 0.13}$ | $\mathbf{0.69_{\pm 0.19}}$ |
| $\ell$ | $0.61_{\pm 0.30}$ | $0.54_{\pm 0.35}$ | $0.17_{\pm 0.15}$ | $\mathbf{0.66_{\pm 0.12}}$ |

the data region. Appendix E.3 provides extended results on additional characters and an ablation study on replacing the contactomorphism ensemble with a single map.

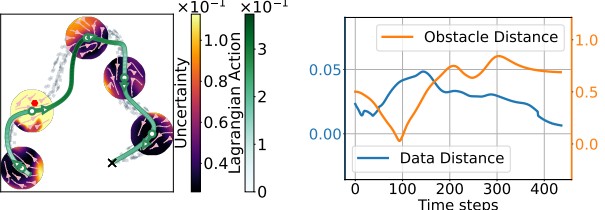

Figure 7. Obstacle avoidance by LEAF_2 (red dot as obstacle).

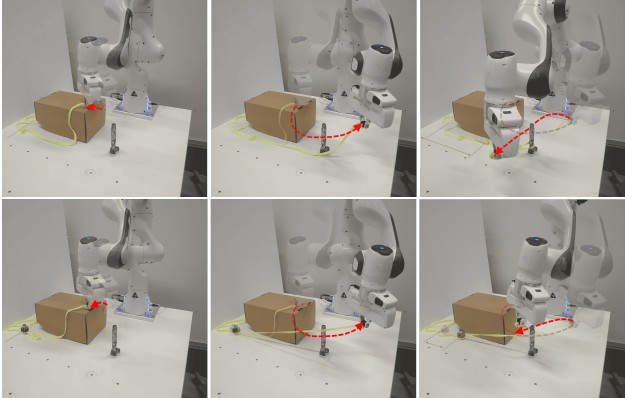

Figure 8. Snapshots of the unloaded (top) and loaded (bottom) WRAP-AND-PULL robotic task, showing rope pickup, wrapping around a drum, and pulling as the robot moves forward.

**WRAP-AND-PULL Robotic Task.** In robotics, learning interaction tasks from demonstrations is more challenging than learning free-space motions (Scherzinger et al., 2019; Le et al., 2021). GCF's ability to model energy behaviors and flexibly generalize to unforeseen states, makes it well-suited for online robot control in interaction tasks. We apply GCF to two robotics tasks: A WRAP-AND-PULL task (detailed below) and a realistic DISHWASHER LOADING task (Simmoteit et al., 2025) (See App. E.4). The first task, a proof of concept, showcases all the robot capabilities enabled by the framework. Three kinesthetic demonstrations were recorded for training, in which the robot grasps a rope, wraps it around a drum, and pulls it along a specific trajectory (see Fig. 8). The ensemble generates a reference dynamic trajectory using the measured robot state, which is tracked by a low-level Cartesian impedance controller. Setup details are in App. D.1.

Our method is implemented using two different latent dynamics: The stable (7b) and the safe (7c) formulations. Table 5 reports the reproduction accuracy for all methods across five trials. The results indicate that the different latent dynamics do not affect the GCF's ability to learn and reproduce the task. In contrast, EF, NCDS and DHNN fail to achieve satisfactory results. EF struggles to model the wrapping phase of the task, while DHNN lacks robustness to

Table 5. Reproduction error (DTWD) in WRAP-AND-PULL task.

| EF | NCDS | DHNN | GCF Safe | GCF Stable |
|---|---|---|---|---|
| $1.79_{\pm 0.04}$ | $3.37_{\pm 0.12}$ | $4.25_{\pm 0.68}$ | $\mathbf{0.62}_{\pm \mathbf{0.22}}$ | $\mathbf{0.61}_{\pm \mathbf{0.25}}$ |

Table 6. Energy Consumption (J) in the WRAP-AND-PULL Task: Comparison of baselines with stable and safe GCF. EF, NCDS and DHNN values are included despite task failure.

| Scenario | EF* | NCDS* | DHNN* | GCF Safe | GCF St. |
|---|---|---|---|---|---|
| Unloaded | $0.54_{\pm .04}$ | $1.52_{\pm .18}$ | $1.01_{\pm .42}$ | $0.60_{\pm .07}$ | $0.59_{\pm .07}$ |
| Loaded | $0.55_{\pm .06}$ | $2.73_{\pm .43}$ | $2.83_{\pm .58}$ | $0.63_{\pm .02}$ | $0.81_{\pm .09}$ |
| $\Delta$ | $0.01_{\pm .07}$ | $1.21_{\pm .47}$ | $1.82_{\pm .72}$ | $\mathbf{0.03}_{\pm .\mathbf{07}}$ | $0.22_{\pm .11}$ |

handle deviations from training states. We test GCF's generalization to unseen energy exchanges by adding a load to the WRAP-AND-PULL task. Table 6 shows energy expenditure for both loaded and unloaded cases. The GCF implementation with stable latent dynamics reproduces the trajectory learned in the unloaded scenario, even in the loaded case, by consuming more energy. In contrast, the safe GCF variant halts motion when the Lagrangian action $s$ approaches zero. As shown in Fig.9, the added load increases energy demands, causing $s$ to vanish before task completion. This triggers an early stop, preventing energy use beyond what was observed during training. This mechanism enables energy-based task completion and acts as a safety stop in the presence of unexpected interactions, as further demonstrated in the DISHWASHER LOADING task. For details on that task and additional analysis of the robot's robustness to physical disturbances in this proof-of-concept, see App.E.4. A full video of the robot experiments, including adaptability to unseen obstacles, and the repository implementing the GCF framework are available at https://sites.google.com/view/geometric-contact-flows.

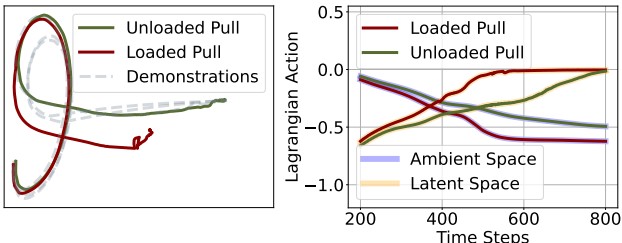

Figure 9. Position and Lagrangian action trajectories (*left* and *right* plots) of the WRAP-AND-PULL task, for both the unloaded and loaded scenarios, using safe GCF.

## 7. Conclusion

Geometric Contact Flows (GCF) model dynamical systems leveraging geometric inductive biases. It starts with a latent dynamics and uses contactomorphisms to adapt it while preserving its key features. To assess uncertainty, GCF employs an ensemble of contactomorphisms to ensure convergence to the data support. GCF has been successfully applied to both dynamics reconstruction and control tasks, covering a wide range of problems including spring mesh simulation, quantum dynamics and robotic manipulation.

## Impact Statement

This paper presents work whose goal is to advance the field of Machine Learning. There are many potential societal consequences of our work, none which we feel must be specifically highlighted here.

## Acknowledgments

The authors are grateful to Hadi Beik-Mohammadi and Huy Le for the insightful discussions on the core technical challenges and prior work in the field, and for the valuable assistance with the robot setup during the experiments.

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

# A. Extended Preliminaries

Hamiltonian and contact Hamiltonian dynamics can be elegantly analyzed via differential geometry. We begin by introducing the symplectic structure and comparing it with the more familiar Riemannian structure. Subsequently, the symplectic geometry is extended to contact geometry, which forms the foundation of our approach. While contact and Riemannian geometries offer distinct perspectives, they are complementary in providing a deeper understanding of the geometry of dynamical systems.

## A.1. Symplectic and Riemannian Structures

Let $\mathcal{M}$ be a smooth compact manifold, and let $\mathcal{T}_x\mathcal{M}$ denote the tangent space at $x \in \mathcal{M}$. The collection of all the tangent spaces identifies the tangent bundle $\mathcal{T}\mathcal{M} = \cup_{x\in\mathcal{M}}\mathcal{T}_x\mathcal{M}$. A vector field $X : \mathcal{M} \to \mathcal{T}\mathcal{M}$ assigns a tangent vector $v$ to each point $x \in \mathcal{M}$. The set of all the vector fields over $\mathcal{T}\mathcal{M}$ is denoted as $\Gamma(\mathcal{T}\mathcal{M})$. A differential 1-form $\alpha : \mathcal{T}\mathcal{M} \to \mathbb{R}$ is a smooth map field acting on vectors of the tangent bundle. For a smooth function $f : \mathcal{M} \to \mathbb{R}$, the 1-form $\alpha = df$ generalizes the gradient from Euclidean spaces. Specifically, $df$ measures the variation of $f$ under an infinitesimal displacement on $\mathcal{M}$. This displacement is locally described by a starting point $x$ and a direction $v$, such that $(x, v) \in \mathcal{T}\mathcal{M}$. Alternatively, it can be globally expressed by a vector field $X$. The variation of $f$ along the vector field $X$ is given by $df(X)$. This variation is independent of the choice of reference frame. To preserve this invariance, $df$ must transform covariantly with $X$. Consequently, the 1-form $\alpha = df$ resides in the cotangent bundle $\mathcal{T}^*\mathcal{M}$, the dual space to $\mathcal{T}\mathcal{M}$. The symplectic and Riemannian structures provide two distinct mechanisms for associating a 1-form to a vector field, thereby establishing connections between the tangent and cotangent bundles. By considering the dynamics governed by the vector field and the scalar function defining the 1-form, a relationship between these elements emerges, as illustrated in Figure 10.

**The Riemannian Metric** A Riemannian metric $g : \mathcal{T}\mathcal{M} \times \mathcal{T}\mathcal{M} \to \mathbb{R}$ is a smooth, symmetric, and positive-definite bilinear field of maps defined on pairs of vectors in the tangent bundle. This enables the introduction of an inner product on the tangent spaces of the manifold, allowing us to measure distances and curve lengths. For a smooth curve $x(t) : [t_0, t_1] \to \mathcal{M}$, the length $l$ w.r.t the metric $g$ is $l = \int_{t_0}^{t_1} \sqrt{g(\dot{x}(t), \dot{x}(t))}dt$, where $\dot{x}(t) \in \mathcal{T}_{x(t)}\mathcal{M}$ is the vector tangent to the curve at $x(t)$. The curve minimizing this length between two points $x(t_0)$ and $x(t_1)$ on $\mathcal{M}$ is called a *geodesic*. Geodesics generalize straight lines in Euclidean space to curved spaces, representing the shortest paths in the geometry induced by $g$. The geometric structure $g(\dot{x}(t), \dot{x}(t))$ does not need to be symmetric with respect to $\dot{x}$ to measure curve lengths.

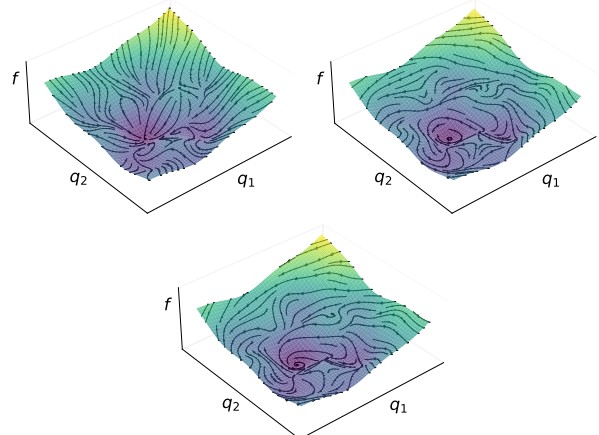

*Figure 10.* The same scalar function $f$, associated with the 1-form $\alpha = df$, gives rise to three distinct vector fields under different geometric structures: Riemannian (top left), symplectic (top right), and contact (bottom). The streamlines of these vector fields are illustrated on a representation of the state manifold. In Riemannian geometry, the streamlines correspond to gradient flow trajectories, following the direction of steepest ascent or descent of $f$, which acts as a potential. In symplectic geometry, the streamlines are instead tangent to the level curves of $f$, representing isoenergetic trajectories where $f$ remains constant, thus describing the dynamics of conservative systems. In contrast, in contact geometry, a single flow line can traverse different energy levels, with the variation of energy along the flow governed by Equation 8.

In such cases, it defines a Finslerian structure, which is a generalization of the Riemannian structure (Chern, 1996).

The Riemannian metric also establishes a correspondence between vector fields and covector fields, thus defining a bijection between the tangent bundle $\mathcal{T}\mathcal{M}$ and the cotangent bundle $\mathcal{T}^*\mathcal{M}$. Specifically, the metric $g$ associates a vector field $X_f$ with the differential 1-form $\alpha = df$ through $df(X) = g(X_f, X), \forall X \in \Gamma(\mathcal{T}\mathcal{M})$. Since $g$ defines an inner product, this connection allows the variation $df(X)$ to be interpreted as a measure of alignment between the vector fields $X$ and $X_f$. A greater alignment corresponds to a larger variation of $f$ along $X$. Consequently, $X_f$ is orthogonal to the level sets of $f$ and represents a gradient vector field. In this sense, $f$ can be interpreted as a potential function, generating a gradient flow $x(t)$ whose trajectories follow geodesic curves with respect to the metric $g$.

**The Symplectic Form** A differential 2-form $\omega : \mathcal{T}\mathcal{M} \times \mathcal{T}\mathcal{M} \to \mathbb{R}$ is a skew-symmetric, bilinear, and smooth field of maps acting on pairs of tangent vectors. A 2-form is called symplectic if it is both closed ($d\omega = 0$) and non-degenerate. Unlike the Riemannian metric, the symplectic form lacks the properties required to define an inner product. However, it still establishes a fundamental relation between differential 1-forms and vector fields: Given a 1-form $df$, the symplectic form $\omega$ uniquely determines a vector field $X_f$

that is tangent to the level sets of $f$, rather than perpendicular as in the Riemannian case. This relation is defined by,

$$df(X) = \omega(X_f, X), \quad \forall X \in \Gamma(\mathcal{T}\mathcal{M}). \quad (20)$$

By definition, $f$ remains constant along the flow of $X_f$, which in turn preserves the symplectic form $\omega$, i.e., $\mathcal{L}_{X_f}\omega = 0$ where $\mathcal{L}_{X_f}$ denotes the Lie derivative (Silva, 2001). In this framework, the function $f$ is interpreted as a conserved energy, or equivalently, as a Hamiltonian $H$. The symplectic structure thereby endows $\mathcal{M}$ with a natural geometric framework for formulating Hamiltonian dynamics (Tokasi & Pickl, 2022). The pair $(\mathcal{M}, \omega)$ is referred to as a *symplectic manifold*. Notably, the non-degeneracy of $\omega$ implies that $\mathcal{M}$ must be even-dimensional.

A diffeomorphism $\phi : (\mathcal{M}, \omega) \to (\mathcal{N}, \omega')$ between two symplectic manifolds is called a *symplectomorphism* if it preserves the symplectic form, i.e., $\phi^*\omega' = \omega$ (Polterovich, 2012). The symplectic flow $x(t)$, generated by integrating the Hamiltonian vector field $X_f$, can be understood as the action of a one-parameter group of diffeomorphisms $\phi(t)$ on a point $x \in (\mathcal{M}, \omega)$. This is defined as,

$$x(t) = \phi(t)(x) : [0, T] \subset \mathbb{R} \to (\mathcal{M}, \omega). \quad (21)$$

### A.2. Symplectic Bundles of Riemannian Manifolds

The geodesic flow $x(t)$ on a Riemannian manifold $(\mathcal{R}, g)$ lifts to the joint evolution of coordinates $(x(t), \alpha(x(t), \dot{x}(t))$ on the cotangent bundle $\mathcal{T}^*\mathcal{R}$ (Abraham & Marsden, 2008). This extended dynamics is governed by an energy function $H(x, \alpha) : \mathcal{T}^*\mathcal{R} \to \mathbb{R} = g^{-1}(\alpha, \alpha)$, which remains constant along the flow. A reparameterization $ds = \sqrt{H}dt$ links the trajectory of the integrated dynamical system at time $t$ on $\mathcal{T}^*\mathcal{R}$ with the length of the corresponding geodesic on $\mathcal{R}$. This framework reveals a fundamental connection between geodesic flows and Hamiltonian dynamics in the special case where the Hamiltonian consists solely of a kinetic energy term. The cotangent bundle $\mathcal{T}^*\mathcal{R}$ is naturally equipped with a symplectic structure, making it a symplectic manifold $(\mathcal{T}^*\mathcal{R}, \omega)$.

The formulation can be further generalized by introducing a potential energy function into the Hamiltonian, given by $H(x, \alpha) = g^{-1}(\alpha, \alpha) + V(x)$. In this setting, the geodesic structure underlying the Hamiltonian flow is determined by the Jacobi metric:

$$\hat{g} = (H - V(x)) \, g, \quad (22)$$

which rescales the original metric $g$ by a position-dependent conformal factor (Abraham & Marsden, 2008). The corresponding time reparameterization takes the form

$$ds = \sqrt{H - V(x)} \, dt, \quad (23)$$

restoring the interpretation of the trajectory as a geodesic with respect to the metric $\hat{g}$ (Udrişte, 2000; Udriste & Udriste, 2000). This reparameterization underlies the Maupertuis principle, which states that conservative mechanical motion can be recast as geodesic motion in a suitably deformed geometry.

**The Maupertuis' principle** The trajectories of an Hamiltonian dynamics on the symplectic manifold $(\mathcal{T}^*\mathcal{R}, \omega)$ are obtained as the critical solutions of the Lagrangian action functional (López & Martínez, 2000),

$$s = \int_{t_0}^{t_1} \alpha(t) \cdot \dot{x}(t) - H(x(t), \alpha(t)) \, dt, \quad (24)$$

where the covector $\alpha(x(t), \dot{x}(t))$ is denoted simply as $\alpha(t)$ to emphasize its dependence on the integration variable $t$. The resolution of this optimization problem corresponds to a geodesic computation on $(\mathcal{M}, \hat{g})$ under the arc-length parametrization of Equation (23),

$$x(s) = \underset{\dot{x}}{\arg\min} \int_{s_0}^{s_1} \sqrt{\hat{g}(\dot{x}(s), \dot{x}(s))} \, ds. \quad (25)$$

A modification of the Hamiltonian dynamics on the symplectic manifold $(\mathcal{T}^*\mathcal{R}, \omega)$ can be represented in the time domain as a change in the action functional (24), while geometrically, it corresponds to an adjustment of the Jacobi metric on the Riemannian manifold $(\mathcal{R}, \hat{g})$. This geometric perspective, first introduced in classical mechanics in the 18th century, is known as the *Maupertuis' principle of least action*. In physical terms, it states that the dynamics of a system follow a trajectory that extremizes (typically minimizes) the difference between the kinetic and potential terms over time. Practically, this provides a geometric framework for shaping or controlling the trajectories of a dynamical system by modifying only the underlying Jacobi metric $\hat{g}$ (Udrişte, 2003; López & Martínez, 2000).

### A.3. Contact Geometry

While symplectic manifolds provide a geometric framework for modeling the dynamics of conservative systems in classical mechanics, a more general approach is required to describe non-conservative systems. This is addressed by contact manifolds, the odd-dimensional counterparts of symplectic manifolds (Geiges, 2001; Bravetti et al., 2017). A *contact manifold* is defined as $(\mathcal{M}, \eta)$, where $\mathcal{M}$ is an odd-dimensional smooth manifold, and $\eta$ is a non-degenerate 1-form known as the contact form (Geiges, 2008). The contact form satisfies the *maximal non-integrability* condition, meaning that the top-degree differential form $\eta \wedge (d\eta)^d \neq 0$ is nowhere vanishing on $\mathcal{M}$. This form is constructed by taking the exterior product of $\eta$ with the $d$-fold wedge product of its exterior derivative $d\eta$, i.e.,

$$(d\eta)^d = \underbrace{d\eta \wedge \cdots \wedge d\eta}_{d \text{ times}}. \quad (26)$$

The $(2d+1)$-form defines a volume form on $\mathcal{M}$, ensuring that the hyperplanes defined by $\ker(\eta) \subset T\mathcal{M}$, which constrain the dynamics on the contact manifold, do not form a foliation, i.e., they do not partition the manifold into lower-dimensional submanifolds (Geiges, 2001; 2008). Geometrically, this means that the contact distribution imposes *non-holonomic constraints*: it restricts the admissible directions of motion at each point without confining the dynamics to a fixed submanifold or energy level. This property is crucial for modeling systems where energy can change over time, enabling constraints on energy behavior without enforcing conservation.

**Contact Hamiltonian Dynamics** Like symplectic geometry, contact geometry connects scalar functions to vector fields, enabling the description of dynamical systems (Zadra, 2023). Given an energy function $H : \mathcal{M} \to \mathbb{R}$, the dynamics on a contact manifold are defined by a contact Hamiltonian vector field $X_H$, as follows,

$$dH(X) = d\eta(X_H, X) - \mathcal{L}_{X_H}\eta(X), \ \forall X \in \Gamma(T\mathcal{M}). \tag{27}$$

Unlike symplectic geometry, where dynamics are confined to energy-preserving flows along the level sets of the Hamiltonian, contact geometry allows for an additional component of motion. Specifically, the dynamics on a contact manifold are not restricted to the term $d\eta(X_H, X)$, which lies tangent to the level sets of $H$, but also include a transverse component $\mathcal{L}_{X_H}\eta(X)$, arising from the non-degeneracy of the contact form. Consequently, while in symplectic geometry the symplectic form $\omega$ is strictly preserved, contact geometry allows the contact form $\eta$ to be preserved only up to a scaling factor $a \in \mathbb{R}$ (Bravetti et al., 2017).

A diffeomorphism $\varphi : (\mathcal{M}, \eta) \to (\mathcal{N}, \eta')$ between two contact manifolds, a.k.a. *contactomorphism*, satisfies $\varphi^*\eta' = a\eta$, thereby preserving the contact structure up to the scaling factor (Zadra, 2023). Analogous to the symplectic case, the integral flow $x(t)$ generated by the contact Hamiltonian vector field $X_H$ can be viewed as a contactomorphism, i.e.,

$$x(t) = \varphi(t)(x) : [0, T] \subset \mathbb{R} \to (\mathcal{M}, \eta). \tag{28}$$

**Contact Flows as Riemannian Geodesics** Contact Hamiltonian flows can also be reinterpreted as geodesics associated with an induced Riemannian metric $\hat{g}$, under the reparameterization given in Equation (23) (Udriște, 2000) In contrast to the symplectic case, contact dynamics are inherently non-conservative: the Hamiltonian function $H(t)$ varies with time, introducing an additional degree of freedom. Let us denote the contact manifold $(\mathcal{T}^*\mathcal{R} \times \mathbb{R}, \eta)$ as the augmented cotangent bundle of a Riemannian manifold $(\mathcal{R}, \hat{g})$. In this setting, contact dynamics corresponds not to geodesic motion directly on $\mathcal{R}$, but rather on an augmented state-time manifold $\mathcal{R} \times \mathbb{R}$. The projection of these geodesics onto $\mathcal{R}$ yields a family of trajectories, each corresponding to a

specific time-dependent energy profile $H(t)$ (Di Cairano et al., 2019). The additional variable defined on $\mathbb{R}$ is the Lagrangian action $s$, introduced in Equation (24), which represents the geodesic length. In the contact setting, the Hamiltonian function also depends on $s$, making the Lagrangian action functional implicit:

$$s = \int_{t_0}^{t_1} \alpha(t) \cdot \dot{x}(t) - H(x(t), \alpha(t), s) \, dt. \tag{29}$$

This dependence renders the associated Jacobi metric $s$–dependent, thereby transforming the geodesic computation problem into:

$$x(s) = \underset{\dot{x}}{\arg\min} \int_{s_0}^{s_1} \sqrt{\hat{g}(\dot{x}(s), \dot{x}(s), s)} \, ds. \tag{30}$$

Thus, the length $s$ of the geodesic on $\mathcal{R}$ directly influences the metric $\hat{g}$, reflecting the non-conservative nature of contact dynamics. The proposed Geometric Contact Flows build on this foundation, modeling the dynamics by harnessing the contact interpretation on the full state manifold while generalizing it through the Riemannian perspective on the configuration manifold.

# B. Methodology Details

This section presents the training algorithms for the single contactomorphism and ensemble approaches. We also outline the integration method for a contact Hamiltonian dynamics, essential to implementing contactomorphism networks. Finally, we examine the implications of preserving the contact structure for the physical interpretability of the ambient dynamics.

## B.1. Training Algorithms

---

**Algorithm 1** Training a Contactomorphism $\varphi_r(T)$

---

**Input:** Dataset: $\{\bar{\mathbf{x}}_b(t)\}_{b\in[1,B]}^{t\in[0,t_b]}$
**Output:** A trained contactomorphism $\varphi_r(T)$
1: **for** $e = 1$ to $E$ **do**     ▷ Iterate over epochs
2:    **for** $b = 1$ to $B$ **do**     ▷ Iterate over trajectories
3:       $\{\bar{\mathbf{z}}_b(t)\}_{t\in[0,t_b]} = \varphi_r(T)\left(\{\bar{\mathbf{x}}_b(t)\}_{t\in[0,t_b]}\right);$
         ▷ Map the trajectories into the latent space using (9)
4:       $\{\mathbf{z}_b(t)\}_{t\in[0,t_b]} = \varphi_g(t)\left(\{\bar{\mathbf{z}}_b(0)\}\right);$
         ▷ Integrate the latent dynamics according to flow (6)
5:       $\{\mathbf{x}_b(t)\}_{t\in[0,t_b]} = \varphi_r^{-1}(T)\left(\{\mathbf{z}_b(t)\}_{t\in[0,t_b]}\right);$
              ▷ Map back to the ambient space with (9)
6:       $\arg\min \ell(\mathbf{x}_b(t), \bar{\mathbf{x}}_b(t), \mathbf{z}_b(t), \bar{\mathbf{z}}_b(t));$
              ▷ Update $\varphi_r(T)$ by minimizing loss (12)
7:    **end for**
8: **end for**

---

**Algorithm 2** Training the Ensemble $\{\varphi_{r^n}(T)\}_{n\in[1,N]}$

---

**Input:** Dataset: $\{\bar{\mathbf{x}}_b(t)\}_{b\in[1,B]}^{t\in[0,t_b]}$
**Output:** A trained ensemble $\{\varphi_{r^n}(T)\}_{n\in[1,N]}$
1: **for** $e = 1$ to $E$ **do**     ▷ Iterate over epochs
2:    **for** $b = 1$ to $B$ **do**     ▷ Iterate over trajectories
3:       **for** $n = 1$ to $N$ **do**
4:          $\{\bar{\mathbf{z}}_b^n(t)\}_{t\in[0,t_b]} = \varphi_{r^n}(T)\left(\{\bar{\mathbf{x}}_b(t)\}_{t\in[0,t_b]}\right)$
         ▷ Map the trajectories into the latent space using (14a)
5:       **end for**
6:       $\{\mathbf{z}_b^n(t)\}_{t\in[0,t_b]} = \varphi_g(t)\left(\{\bar{\mathbf{z}}_b^n(0)\}\right);$
         ▷ Integrate the latent dynamics according to flow (6)
7:       **for** $j \sim \text{RandomShuffle}(1, 2, \ldots, N)$ **do**
8:          $\{\mathbf{x}_b^j(t)\}_{t\in[0,t_b]} = \varphi_{r^j}^{-1}(T)\left(\{\mathbf{z}_b^n(t)\}_{t\in[0,t_b]}\right)$
              ▷ Map back to the ambient space with (14b)
9:       **end for**
10:      $\arg\min \frac{1}{N}\sum_{n,j=1}^{N} \ell(\mathbf{x}_b^j(t), \bar{\mathbf{x}}_b(t), \mathbf{z}_b^n(t), \bar{\mathbf{z}}_b^n(t));$
              ▷ Update the ensemble by minimizing loss (12)
11:   **end for**
12: **end for**

---

## B.2. Integrating a Contact Hamiltonian Flow

A contact Hamiltonian dynamic system can be defined using a contact Hamiltonian function $H$ on a contact manifold $(\mathcal{M}, \eta)$. The contact form $\eta$ establishes a connection between $H$ and its associated vector field $X_H$ via Equation (2). In local coordinates, the vector field $X_H$ is expressed as,

$$X_H : \begin{cases} \dot{q}_i = \frac{\partial H}{\partial p_i}, \\ \dot{p}_i = -\frac{\partial H}{\partial q_i} - p_i\frac{\partial H}{\partial s}, \\ \dot{s} = p_i\frac{\partial H}{\partial p_i} - H. \end{cases} \tag{31}$$

This vector field generates a flow $\varphi$ satisfying

$$\frac{d}{dt}\varphi_t(x) = X_H(\varphi_t(x)), \quad \varphi_0(x) = x. \tag{32}$$

The flow preserves the contact form $\eta$ up to a scaling factor $a \in \mathbb{R}$, such that $\varphi^*\eta' = a\eta$. Therefore, the resulting transformation $x(t) = \varphi(t)(x_0)$ is formally a contactomorphism within $(\mathcal{M}, \eta)$. However, the numerical integration of this flow can introduce errors, depending on the method used. Contact splitting integrators are a family of numerical algorithms to integrate contact Hamiltonian flows while preserving the contact structure (Zadra, 2023). This algorithm applies to Hamiltonian functions that can be expressed as a sum of separate terms, with the state $\{\mathbf{q}, \mathbf{p}, s\}$ updated in a corresponding number of steps. In our case, the Hamiltonian function (11) of the individual contactomorphism $\varphi_{r_{\theta_k}}$ consists of three terms, resulting in the following three-step update process,

$$1. \begin{cases} \mathbf{q}_{j+1} = \mathbf{q}_j, \\ \mathbf{p}_{j+1} = \mathbf{p}_j - (\mathbf{p}_j F_j + \nabla F_j s_j)\tau, \\ s_{j+1} = s_j - F_j s_j \tau, \end{cases} \tag{33a}$$

$$2. \begin{cases} \mathbf{q}_{j+2} = \mathbf{q}_{j+1}, \\ \mathbf{p}_{j+2} = \mathbf{p}_{j+1} - \nabla V_{j+1}\tau, \\ s_{j+2} = s_{j+1} - V_{j+1}\tau, \end{cases} \tag{33b}$$

$$3. \begin{cases} \mathbf{q}_{j+3} = \mathbf{q}_{j+2} + (M_{j+2} + \mathbf{p}_{j+2}^\top\nabla M_{j+2})\mathbf{p}_{j+2}\tau, \\ \mathbf{p}_{j+3} = \mathbf{p}_{j+2}, \\ s_{j+3} = s_{j+2} + \mathbf{p}_{j+2}^\top(\mathbf{p}_{j+2}^\top\nabla M_{j+2} + M_{j+2})\mathbf{p}_{j+2}\tau, \end{cases}$$
$$\tag{33c}$$

where $M_j = M(\mathbf{p}_j)$, $V_j = V(\mathbf{q}_j)$, and $F_j = F(\mathbf{q}_j)$. This integration method has the key advantage of being analytically invertible. As a result, a contactomorphism implemented according to this scheme can be efficiently reversed with minimal computational cost.

## B.3. Approximation Error of the Contactomorphism

We experimentally evaluate the numerical error introduced by our implementation of the mapping between the ambient

and latent states by leveraging the analytical contact transformation conditions established in (Bravetti et al., 2017),

$$p_i \frac{d\hat{s}}{ds} - p_i \hat{p}_i \frac{d\hat{q}_i}{ds} = -\frac{d\hat{s}}{dq_i} + \hat{p}_i \frac{d\hat{q}_i}{dq_i}, \qquad (34a)$$

$$\frac{d\hat{s}}{dp_i} - \hat{p}_i \frac{d\hat{q}_i}{dp_i} = 0. \qquad (34b)$$

Here, the coordinates $(q_i, p_i, s)$ and $(\hat{q}_i, \hat{p}_i, \hat{s})$ represent the variables before and after the contact transformation, respectively. The partial derivatives correspond to entries in the Jacobian of the contactomorphism. By computing the difference between the left-hand and right-hand sides of the conditions above, we consistently observe a numerical error lower than $1e-5$, confirming that the implemented transformation closely satisfies the contactomorphism conditions.

### B.4. Physical Interpretability

Equipping the latent dynamics with a contactomorphism allows us to model a dynamical system directly in ambient coordinates. To predict the ambient trajectory from an initial point $\mathbf{x}_0$, we first map it to the latent space via the learned contactomorphism, $\mathbf{z}_0 = \varphi_r(T)(\mathbf{x}_0)$. We then integrate the latent dynamics to obtain $\mathbf{z}(t) = \varphi_g(t)(\mathbf{z}_0)$, and finally map the result back to the ambient space using the inverse contactomorphism, $\mathbf{x}(t) = \varphi_r^{-1}(T)(\mathbf{z}(t))$. Formally, this process is expressed as a composition of contact Hamiltonian flows,

$$\mathbf{x}(t) = \varphi_r^{-1}(T) \circ \varphi_g(t) \circ \varphi_r(T)(\mathbf{x}_0), \ t \in [0, t_1]. \quad (35)$$

Note that the composition of Hamiltonian paths is itself a Hamiltonian path. The corresponding Hamiltonian function, associated with the Riemannian metric $\hat{m}$, can be derived using (Polterovich, 2012, Proposition 1.4.D),

$$H_m = H_r + H_g(\varphi_r^{-1}) - H_r(\varphi_g^{-1}), \qquad (36)$$

where the Hamiltonian of the contactomorphism $H_r$ is,

$$H_r = H_{r_1} + \sum_{k=1}^{K} H_{r_k}(\varphi_{r_{k-1}} \circ \cdots \circ \varphi_{r_1}). \qquad (37)$$

Preserving the contact structure ensures that the predicted ambient dynamics follow a contact Hamiltonian form, making them physically interpretable. Specifically, using a contactomorphism instead of a generic diffeomorphism allows for the derivation of an analytical expression for the Hamiltonian function in the ambient space. The damping coefficient, $\partial H_m / \partial s$, embedded in the contact form, quantifies energy dissipation or absorption as described by Equation (8), and governs the interplay between conjugate variables. While GCF morphs the latent contact form $\eta'$ and modulates the damping coefficient, it also preserves their relationship, ensuring physical consistency. Figure 1 shows that GCF yields a damping coefficient consistent with the real system demonstrating its robustness in modeling conjugate variable interactions.

## C. Implementation Details

### C.1. Importance of Contact Structure Preservation

The transformation $\varphi_r : (\mathcal{M}, \eta) \to (\mathcal{N}, \eta')$ is designed to preserve the contact structure, ensuring that the physical relationships between conjugate variables in the latent baseline dynamics are conserved. The theoretical justification for this is provided in Section 3 and Appendix A B.4. Here, we empirically evaluate its importance by examining the consequences of replacing contactomorphisms by standard diffeomorphisms. This substitution disrupts the underlying physical coherence, leading to degraded reconstruction and generalization performance. Figure 11 qualitatively illustrates the impact of replacing contactomorphisms by diffeomorphisms, highlighting distortions in both the reconstruction and generalization of the LEAF_2 character from the LASA dataset. Figure 12 shows a more elaborated comparison of generalization performance, by considering predicted trajectories initialized from a grid of points in the state space. The use of contactomorphisms shows significantly faster convergence toward the data manifold, measured as the percentage of time steps spent near the demonstrations over the trajectory's duration. Tables 7 and 8 quantitatively confirm the degradation in reconstruction and generalization accuracy when contact structure preservation is not enforced.

*Table 7.* Reconstruction error via DTWD for the ablation study on the structure of the ambient space–latent space mapping

| Character | Contactomorphism | Diffeomorphism |
|---|---|---|
| ⟨ | $\mathbf{0.44_{\pm 0.12}}$ | $1.15_{\pm 0.64}$ |
| ∽ | $\mathbf{0.43_{\pm 0.07}}$ | $0.96_{\pm 0.49}$ |

*Table 8.* Generalization measured as average ratio of convergence for the ablation study on the structure of the mapping between the ambient space and the latent space

| Character | Contactomorphism | Diffeomorphism |
|---|---|---|
| ⟨ | $\mathbf{0.61_{\pm 0.21}}$ | $0.08_{\pm 0.25}$ |
| ∽ | $\mathbf{0.62_{\pm 0.13}}$ | $0.12_{\pm 0.23}$ |

### C.2. Training Details and Weights Ablation

The weights employed in the loss function (12) are $w_x = 1$ and $w_z = 0.01$. Training is conducted for 5000 epochs, taking approximately 4 hours on average. Details of the machine used for training are provided in Appendix D.1. The initial learning rate is $1 \times 10^{-3}$ and is reduced by a factor of 0.9 on plateaus observed for 200 epochs. The loss is clipped at $1 \times 10^3$, and the gradient is clipped at 0.1. Optimization is performed using the Adam optimizer with

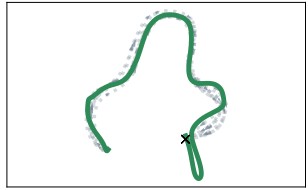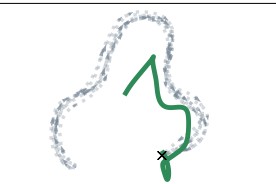

*Figure 11.* This figure illustrates the reconstruction and generalization results for the character LEAF_2 when the contactomorphism connecting the ambient and latent spaces is replaced by a diffeomorphism. As a result, the Hamiltonian structure of the ambient dynamics is lost, removing the physical bias that aids in accurately reconstructing the dynamics. Consequently, the trajectories in the ambient space appear more distorted, especially near the target.

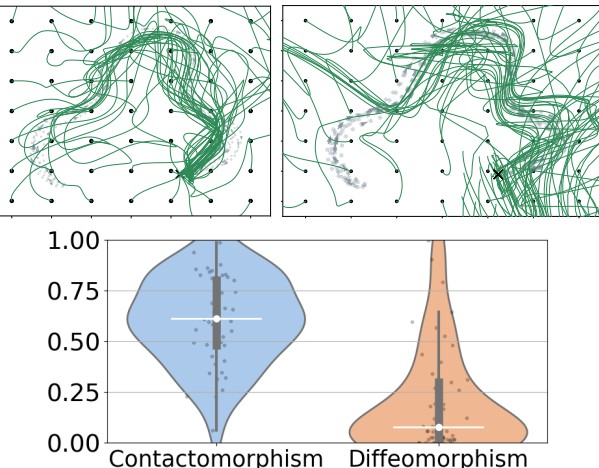

*Figure 12.* The top plots show trajectory predictions from grid points for the character LEAF_2, comparing a model using a single contactomorphism (left) to one using a naive diffeomorphism without physical structure (right). The bottom chart summarizes these trajectories by representing them as points based on the ratio of time steps spent within the data manifold to the total number of time steps. Removing the physical structure disrupts the relationships between state components in the ambient space, negatively impacting generalization.

default hyperparameters, and L2 regularization is applied to the weights with a coefficient of $1 \times 10^{-10}$. While the ambient weighting factor $w_x$ focuses on the reconstruction of the demonstrations in the ambient space, the latent weighting factor $w_z$ plays a crucial role in regularizing the deformation of the latent space.

During training, as detailed in Algorithm 1, the integration of the latent dynamics $z_b(t)$ depends only on the initial reference state $\bar{z}_b(0)$ and remains independent of the rest of the latent demonstrations $\bar{z}_b(t)$. The inverse contactomorphism $\varphi_r^{-1}$ is applied at each point along $z_b(t)$ to recover the corresponding ambient trajectory $\bar{x}_b(t)$, while the forward contactomorphism $\varphi_r$ is used only to map the initial states of the demonstrations to the latent space. As a result, if training relies solely on the ambient loss term, the latent space receives limited structural guidance. Since the contactomorphism defines the mapping from ambient to latent space, this can lead to irregularities or lack of smoothness in the latent representation. In contrast, the latent loss term enforces alignment between the reference trajectory $\bar{z}_b(t)$ and the integrated dynamics $z_b(t)$, thereby promoting a smoother and more coherent latent mapping. This regularization improves the latent space structure, leading to more accurate ambient reconstructions and mitigating unpredictable behavior outside the data manifold. Ultimately, it supports a more stable and generalizable representation. Figure 13 compares the ambient and latent reconstructions of the demonstrations for three different values of $w_z$, illustrating how a small weight can serve as a trade-off to enhance both reconstructions. A quantitative analysis of this ablation study is reported in Table 16.

**C.3. Hamiltonian Function Parametrization**

Each individual parametrized flow $\varphi_{r_{\theta_k}}$, which collectively defines the contactomorphism $\varphi_r$, integrates the vector field associated with the Hamiltonian function in Equation (11). The functions $M_{\theta_k}(\mathbf{p}), V_{\theta_k}(\mathbf{q}), F_{\theta_k}(\mathbf{q})$ are learned compo-

*Table 9.* Reconstruction error via DTWD for weight loss ablation

| $w_z$ | **Ambient Space** | **Latent Space** |
|---|---|---|
| 0 | $0.89_{\pm 0.11}$ | $5.1_{\pm 0.53}$ |
| 0.001 | $0.50_{\pm 0.10}$ | $4.9_{\pm 0.43}$ |
| 0.01 | $\mathbf{0.44_{\pm 0.07}}$ | $1.9_{\pm 0.27}$ |
| 0.1 | $0.56_{\pm 0.06}$ | $1.6_{\pm 0.19}$ |
| 1 | $0.69_{\pm 0.06}$ | $\mathbf{0.96_{\pm 0.15}}$ |

nents representing the inertia, potential energy, and damping coefficient of the Hamiltonian system, respectively. Learning all these components enables the resulting contactomorphism to effectively model a wide range of target dynamics. Table 10 reports reconstruction scores for the character LEAF_2 from the LASA dataset, comparing the performance of the full Hamiltonian with that of simplified versions using fewer parametrized components. The fully parametrized Hamiltonian consistently achieves the best reconstruction accuracy. These functions are implemented using Random Fourier Features Networks (RFFNs), which were selected due to their superior performance in the ablation study reported in Table 11. RFFNs provide a strong balance of expressivity and smoothness, making them particularly well-suited for gradient-based training in this context.

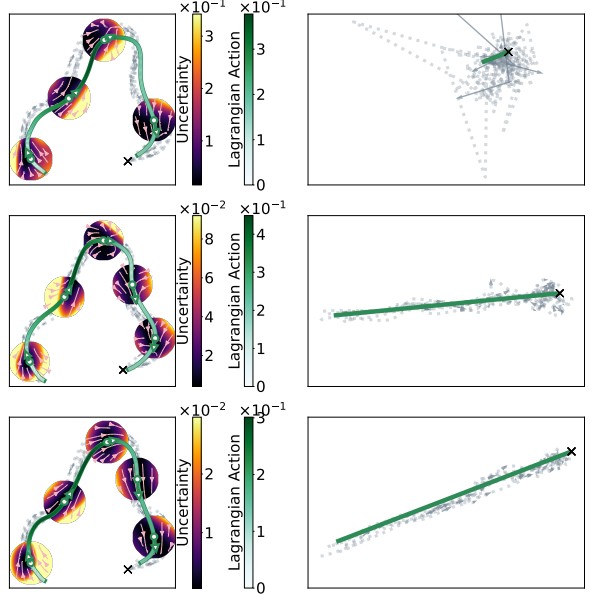

*Figure 13.* Demonstration reconstruction in the ambient space (left) and latent space (right) for different values of the latent weighting factor $w_z$: $w_z = 0$ (top), $w_z = 0.01$ (middle), and $w_z = 1$ (bottom).

*Table 10.* Reconstruction error of character LEAF_2 via DTWD for different parametrization of the contact Hamiltonian

| Hamiltonian | Score |
|---|---|
| $\frac{1}{2}\mathbf{p}^\top M(\mathbf{p})\mathbf{p} + V(\mathbf{q}) + F(\mathbf{q})s$ | $\mathbf{0.44_{\pm 0.12}}$ |
| $\frac{1}{2}\mathbf{p}^\top \mathbf{p} + V(\mathbf{q}) + F(\mathbf{q})s$ | $0.61_{\pm 0.08}$ |
| $\frac{1}{2}\mathbf{p}^\top M(\mathbf{p})\mathbf{p} + V(\mathbf{q})$ | $0.94_{\pm 0.18}$ |
| $\frac{1}{2}\mathbf{p}^\top \mathbf{p} + V(\mathbf{q})$ | $2.14_{\pm 0.52}$ |

*Table 11.* Reconstruction error via DTWD for different learning models used to reconstruct the contact Hamiltonian.

| Character | MLP | RBF | RFFN |
|---|---|---|---|
| ⌣ | $0.94_{\pm 0.18}$ | $0.58_{\pm 0.15}$ | $\mathbf{0.44_{\pm 0.12}}$ |
| ⌇ | $0.69_{\pm 0.11}$ | $0.48_{\pm 0.09}$ | $\mathbf{0.43_{\pm 0.07}}$ |

## D. Experimental Setup

### D.1. System Configuration and Architectural Settings

The framework runs on a machine equipped with 13th Gen Intel Core i7-13850HX CPUs. Positional data is normalized to the range $[-0.5, 0.5]$. Other coordinates are adjusted accordingly to maintain physical coherence. Additionally, velocity data is normalized to $[-1, 1]$ by scaling the duration of the trajectory. In the experiments, the input to our framework is the current system state at $t_k$, while the output is the next state at $t_{k+1}$. This prediction is repeated at each time step to reconstruct the full dynamics. Our approach treats the dynamical system as autonomous, excluding time as an explicit input. The number of contactomorphisms $N$ used in the ensemble depends on the task, whether it involves reconstructing a dynamical system or generalizing a demonstrated trajectory for control execution. In the first case (e.g., the spring mesh and quantum system), a single contactomorphism ($N = 1$) is used. In the second case (e.g., the handwriting dataset and robotics tests), the ensemble consists of $N = 5$ contactomorphisms. In the latent space, the stable baseline dynamics given by Equation (7b) is used in all experiments. Additionally, for the execution of the robotics tasks, the safe dynamics from Equation (7c) is also employed to leverage its energy-aware behavior.

Each contactomorphism $\varphi_r$, is composed of a sequence of $K = 12$ individual parametrized flows $\varphi_{r_{\theta_k}}$, with a duration of $\tau = 0.2$ seconds each. The scalar functions defining each flow, $M_{\theta_k}(\mathbf{p}), V_{\theta_k}(\mathbf{q}), F_{\theta_k}(\mathbf{q})$, are implemented using Random Fourier Features networks with $n_f = 200$ hidden units and a kernel bandwidth of 1. A $\tanh$ activation function bounds the output of these scalar functions to the range $[-2, 2]$. Across experiments, the GCF networks consist of 7200 trainable parameters. The baseline models were evaluated with various parameter configurations, ranging from the original author-released versions to implementations with parameter counts comparable to that of GCF. Ultimately, we selected the architecture that yielded the best performance for each method (EF: 2000, NCDS: 2000, DHNN: 4572, HNN: 2286, MLP: 7168).

### D.2. Robot setup.

The WRAP-AND-PULL and the DISHWASHER LOADING robotics tasks employ a 7-DoF Franka Emika Panda robot as the test platform, with ROS2 acting as the middleware to interface either the GCF or the state-of-the-art benchmark method with the robot. The trajectory predicted by the GCF or the benchmark method is sent to a low-level Cartesian Impedance Controller, which is built on top of the Franka torque control system using libfranka functionalities. Via the Franka Control Interface (FCI), motor signals are sent to the robotic manipulator in real-time at a rate of 1 KHz. The GCF learning model, trained according to the

task demonstrations, is loaded by a control node operating at a frequency of 10 Hz. This node receives state measurements from the robot, normalizes the data, and computes the Lagrangian action before calling the model to predict dynamics for the subsequent 2 seconds. At each timestep, the GCF node checks whether the GCF learning model is available for computation. If the model is not already engaged, it processes the latest robot state. The node then updates the Impedance Controller with a new frame of the predicted dynamics. Whenever a new trajectory becomes available, the previously executed trajectory is replaced and updated accordingly. The control node used for the state-of-the-art benchmarks shares the same features.

### D.3. Conversion to Canonical Coordinates

In classical mechanics, the state of physical systems is typically described using positions and velocities $\{\mathbf{q}, \dot{\mathbf{q}}\}$, rather than the canonical coordinates $\{\mathbf{q}, \mathbf{p}, s\}$, required in the contact Hamiltonian formulation. Therefore, prior to inferring the network, it is necessary to perform a conversion and estimate the action Lagrangian value $s$, which is not directly observable in many applications.

**Spring Mesh System and Handwriting Datasets.** The spring mesh dataset, along with the LASA and DigiLeTs handwriting datasets, represent the system dynamics using the state variables $\mathbf{q}, \dot{\mathbf{q}}$. To transform these data into canonical coordinates, we assume that the dynamical system is characterized by a unitary mass, implying $\dot{q} = p$, and governed by a contact Hamiltonian in mechanical form, defined as,

$$H = \frac{1}{2}\mathbf{p}^\top \mathbf{p} + \beta(\mathbf{q}) + \gamma(\mathbf{q})s, \qquad (38)$$

where $\beta(\mathbf{q})$ is the potential energy, and $\gamma(\mathbf{q})$ denotes the damping coefficient. Depending on its sign, $\gamma(\mathbf{q})$ can account for either energy dissipation ($\gamma > 0$) or energy generation ($\gamma < 0$). In this context, the dynamics are characterized by an attraction point, and we model the potential function during demonstrations as linearly decreasing toward zero. In order to compute the Lagrangian action $s$, the potential energy is parametrized as $\beta(\mathbf{q}) = T - t$, where $T$ represents the final time of the trajectory.

According to the Maupertuis' principle of least action, the variation of $s$ is,

$$\dot{s} = \frac{1}{2}\dot{\mathbf{q}}^\top \dot{\mathbf{q}} - \beta(\mathbf{q}) - \gamma(\mathbf{q})s. \qquad (39)$$

In addition, as derived from Noether (1918), the variation of the contact Hamiltonian function (38) is regulated by,

$$\left(\frac{1}{2}\dot{\mathbf{q}}^\top \dot{\mathbf{q}} + \beta(\mathbf{q}) + \gamma(\mathbf{q})s\right) = H_0 e^{-\int_0^T \gamma(\mathbf{q})dt}. \qquad (40)$$

By substituting Equation (39) into (40), we obtain,

$$\left(\dot{\mathbf{q}}^\top \dot{\mathbf{q}} - \dot{s}\right) = H_0 e^{-\int_0^T \left(\frac{1}{2}\dot{\mathbf{q}}^\top \dot{\mathbf{q}} - \beta(\mathbf{q}) - \dot{s}\right)/s\, dt}. \qquad (41)$$

Differentiating this equation with respect to time, we derive,

$$e^{-\int_0^T \left(\frac{1}{2}\dot{\mathbf{q}}^\top \dot{\mathbf{q}} - \beta(\mathbf{q}) - \dot{s}\right)/s\, dt}\left(\rho(s, \dot{s}, \dot{\mathbf{q}}, \ddot{\mathbf{q}}) - \ddot{s}\right) = 0, \quad (42a)$$

$$\rho(s, \dot{s}, \dot{\mathbf{q}}, \ddot{\mathbf{q}}) = \frac{\frac{1}{2}\dot{\mathbf{q}}^\top \dot{\mathbf{q}} - \beta(\mathbf{q}) - \dot{s}}{s}\left(\dot{\mathbf{q}}^\top \dot{\mathbf{q}} - \dot{s}\right) + 2\dot{\mathbf{q}}^\top \ddot{\mathbf{q}}, \qquad (42b)$$

which implies:

$$\ddot{s} = \frac{\frac{1}{2}\dot{\mathbf{q}}^\top \dot{\mathbf{q}} - \beta(\mathbf{q}) - \dot{s}}{s}\left(\dot{\mathbf{q}}^\top \dot{\mathbf{q}} - \dot{s}\right) + 2\dot{\mathbf{q}}^\top \ddot{\mathbf{q}}. \qquad (43)$$

This differential equation can be integrated forward to obtain the full time series of Lagrangian actions $s$, given the initial conditions,

$$\ddot{s}_0 = 0, \quad \dot{s}_0 = \dot{\mathbf{q}_0}^\top \dot{\mathbf{q}}_0, \quad s_0. \qquad (44)$$

Through this process, we obtain a description of the dynamics in canonical coordinates $\{\mathbf{q}, \mathbf{p}, s\}$.

**Robotic Tasks.** The dynamical system under consideration consists of the Franka-Emika Panda robot and its workspace and manipulated objects, represented by the rope. We have access to the end-effector position and velocity, denoted as $\{\mathbf{q}, \dot{\mathbf{q}}\}$. To transform these variables into canonical coordinates in real time, we first use the dynamical model of the robot to compute the workspace inertia matrix $\Lambda$. Assuming a Hamiltonian in mechanical form, the canonical momentum is obtained as $\mathbf{p} = \Lambda\dot{\mathbf{q}}$. The total energy of the system at a given time $t$ can be modeled as the sum of the robot kinetic energy, the dissipation term, and the energy stored in the environment, which is represented by the displacement of the rope as a result of the work performed by the robot. The total energy is therefore expressed as,

$$H(t) = \frac{1}{2}\dot{\mathbf{q}}^\top \Lambda \dot{\mathbf{q}} + s - \int_0^t \mathbf{f}_e^\top \delta\mathbf{q}\, d\tau, \qquad (45)$$

where $\mathbf{f}_e$ represents the external force applied at the end-effector and $\delta\mathbf{q}$ denotes an infinitesimal robot motion. The integral term $\int_0^t \mathbf{f}_e^\top \delta\mathbf{q}\, d\tau$ is negative during a pulling action and includes both the conservative and non-conservative components of the robot interaction with the environment.

The evolution of the Lagrangian action $s$ is governed by the Maupertuis' principle of least action according to the equation,

$$\dot{s} = \frac{1}{2}\dot{\mathbf{q}}^\top \Lambda \dot{\mathbf{q}} - s + \int_0^t \mathbf{f}_e^\top \delta\mathbf{q}\, d\tau. \qquad (46)$$

This differential equation shows how $s$ can be updated at every time instant according to the measured values of $\dot{\mathbf{q}}$ and $\mathbf{f}_e$. Through this process, we obtain a description of the dynamics in canonical coordinates $\{\mathbf{q}, \mathbf{p}, s\}$.

# E. Extended Results

## E.1. Spring Mesh System

Spring networks are meshes of particles connected by springs, effectively modeling deformable surfaces and solid materials (Pfaff et al., 2021). They serve as simple yet versatile proxies for a wide range of physical scenarios involving deformable solids and cloth, including mechanical simulations of material deformation such as finite element modeling, as well as robotics tasks and computer graphics applications. By benchmarking on a spring mesh, we can examine how Geometric Contact Flows learns large deformations, wave propagation through a membrane, or the impact of damping. Each pair of connected particles $(a, b)$ exchanges spring forces regulated by the Hooke's law, as

$$\mathbf{f}_{ab} = -k \cdot (\|\mathbf{q}_a - \mathbf{q}_b\|_2 - l_{ab}) \, \frac{\mathbf{q}_a - \mathbf{q}_b}{\|\mathbf{q}_a - \mathbf{q}_b\|_2} - \gamma \dot{\mathbf{q}}_a, \quad (47)$$

where $l_{ab}$ is the rest length of the spring, $\gamma$ is the damping coefficient, and $k$ is the stiffness. The coupling of multiple springs leads to complex large-scale deformations and oscillatory behavior.

In this assessment, we use the 60-dimensional dataset originally introduced by Otness et al. (2021), as part of a comprehensive benchmark for learning physical systems. The system comprises a 2D square grid of particles $(4 \times 4)$, connected by springs, with the top row of particles fixed in place. Springs are placed along both the axis-aligned edges and the diagonals of each grid square. The rest lengths of the springs are chosen such that the uniformly spaced particles remain at equilibrium in their initial configuration. Initial conditions are generated by perturbing the positions of all non-fixed particles. These perturbations are drawn as uniform random vectors within a circle of radius $r$, ensuring that all sampled initial conditions have zero initial momentum. The training dataset comprises 20 spring-mesh systems, each characterized by a distinct set of initial conditions. The system parameters are detailed in Table 12. Each system is simulated over a time horizon of 8 seconds, capturing the complete trajectories of particle displacements and momenta. The resulting dataset offers a comprehensive representation of the system's dynamical behavior across the explored parameter space. Figure 14 illustrates the oscillatory behavior of the first three nodes in position space, comparing the true dynamics with the predictions of the GCF approach and baseline models. Quantitative metrics evaluating the accuracy of the dynamics prediction are presented in Table 1. The GCF model achieves a 57% reduction in reconstruction error compared to DHNN, the next-best baseline, owing to the convergence biases integrated into the network architecture.

*Table 12.* Spring Mesh Parameters

| Parameter | Values |
|---|---|
| damping coefficient $\gamma$ [kg/s] | 0.1 |
| mass [kg] | 1 |
| stiffness $k$ [N/m] | 1 |
| initial displacement $r$ [m] | Uniform$(0.1, 0.35)$ |

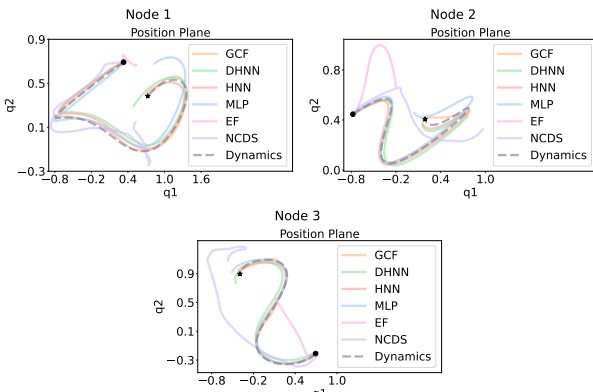

*Figure 14.* Dynamics reconstruction of the first three nodes in the spring mesh experiment. The reference dynamics (dashed gray line) is compared against the predictions from GCF and other baseline models.

## E.2. Quantum Mechanical System

**Contact Hamiltonian formulation of quantum systems.** The versatility of the contact Hamiltonian framework extends its applicability beyond classical mechanics, enabling the description of thermodynamic processes (Bravetti, 2019; Simoes et al., 2020) and certain aspects of quantum mechanical systems (Woit et al., 2017; Herczeg & Waldron, 2018). Although this formulation cannot account for decoherence or other effects characteristic of open quantum systems, it provides a natural extension of the Schrödinger equation for the evolution of pure states in finite-level quantum systems (Ciaglia et al., 2018). Some authors (Cruz, 2018; Bravetti et al., 2017) proposed to use the contact formulation in the context of canonical quantization in position representation. However, this approach has the drawback of not preserving the norm of the wave function in the Schrödinger equation.

In order to facilitate the application of contact geometry in quantum mechanics, we eliminate the detrimental effects of dissipation on the probability distribution by reformulating the Schrödinger equation in terms of the Madelung representation. Starting from the standard Schrödinger equation,

$$i\hbar \frac{\partial \Psi}{\partial t} = \hat{H} \Psi, \quad (48)$$

where $\hbar$ is the reduced Planck constant, $\Psi$ is the wave func-

tion, and the Hamiltonian operator is given by

$$\hat{H} = \frac{\hat{p}^2}{2m} + V(\hat{q}), \qquad (49)$$

expressed in terms of the position and momentum operators, $\hat{q}$ and $\hat{p}$, we rewrite the wave function in polar form as

$$\Psi = \sqrt{\rho}\, e^{is/\hbar}, \qquad (50)$$

where the probability density function, $\rho = |\Psi|^2$, corresponds to the squared amplitude of the wave function, while $s$ represents the phase, related to the classical action. Substituting this expression into the Schrödinger equation and separating real and imaginary components, we obtain two real-valued equations. The first one is the continuity equation,

$$\frac{\partial \rho}{\partial t} + \nabla_q \cdot (\rho \nabla_q s) = 0. \qquad (51)$$

The second one is the quantum Hamilton-Jacobi equation,

$$\frac{\partial s}{\partial t} + \frac{(\nabla_q s)^2}{2m} + V + Q = 0, \qquad (52)$$

where,

$$Q = -\frac{\hbar^2}{2m} \frac{\Delta_q \sqrt{\rho}}{\sqrt{\rho}} \qquad (53)$$

is the quantum potential, with the operator $\Delta_q$ denoting the Laplacian. The Madelung equations are particularly suitable for adaptation within the contact Hamiltonian framework, as the extension of the Hamiltonian operator (49) to the contact Hamiltonian operator,

$$\hat{H} = \frac{\hat{p}^2}{2m} + V(\hat{q}) + \gamma s, \qquad (54)$$

modifies the quantum Hamilton-Jacobi equation (52) to,

$$\frac{\partial s}{\partial t} + \frac{(\nabla_q s)^2}{2m} + V + Q + \gamma s = 0. \qquad (55)$$

This introduces dissipation into the contact system's evolution without altering the properties of the probability distribution, which remain governed by the continuity equation (51).

**Quantum dynamics reconstruction**    In this test, we focus on reconstructing the dynamics of a single-mode bosonic system, whose Hamiltonian consists of a linear term representing the free evolution of the mode, a squeezing-like interaction, and a cubic nonlinearity. Expressed in terms of the position and momentum operators, the contact Hamiltonian is given by,

$$\hat{H} = \frac{1}{2}\omega(\hat{p}^2 + \hat{q}^2) - 2\chi \hat{q}\hat{p} + \frac{2}{3}\beta \hat{q}(\hat{q}^2 - \hat{p}^2) + \gamma s. \quad (56)$$

To account for small fluctuations arising from experimental imperfections, calibration errors, and unavoidable environmental disturbances, we introduce a stochastic perturbation to the system's evolution. Unlike open quantum system approaches, this modification does not lead to decoherence, ensuring that the system remains in a pure state while incorporating realistic uncertainties. The deterministic Schrödinger equation (48) is therefore extended to,

$$i\hbar\, d\Psi = dt\hat{H}\Psi + i\hbar\, \delta\, dW_t\Psi, \qquad (57)$$

where $dW_t$ is a Wiener process representing stochastic fluctuations, and $\delta$ determines their magnitude. In this experiment, 20 dynamical systems have been generated by integrating the Madelung representation of Equation 57 for 8 seconds, i.e.,

$$\frac{\partial \rho}{\partial t} = -\nabla_q \cdot (\rho \nabla_q s) + \delta^2 \frac{\hbar}{2}\Delta_q \rho; \qquad (58a)$$

$$\frac{\partial s}{\partial t} + \frac{(\nabla_q s)^2}{2m} + V + Q + \gamma s = 0. \qquad (58b)$$

The parameters of the quantum system were sampled from uniform distributions, as detailed in Table 13. We evaluate the performance of GCF in reconstructing the dynamics of the expected values of the position and momentum operators, $\hat{q}$ and $\hat{p}$, as well as the phase variable $s$, in comparison to the baselines. Unlike other tests, the HNN and DHNN frameworks cannot be applied to the quantum case, as they operate only in even-dimensional phase spaces and cannot model the additional odd variable $s$. Stochastic results are summarized in Table 2. GCF reduces reconstruction error by 60% in modeling the system's evolution. Figure 15 illustrates an example of a single trajectory, comparing the reconstructions produced by the models.

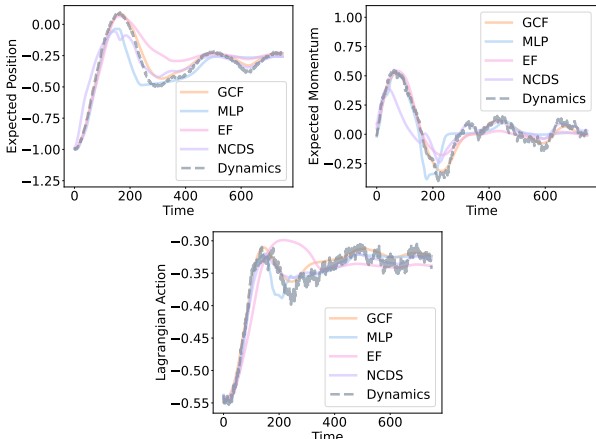

*Figure 15.* Trajectories of position $q$, momentum $p$, and Lagrangian action $s$ for the evolution of a single-phase bosonic system. The trajectory obtained from integrating the Schrödinger equation (gray dashed line) is compared against the predicted evolution from the GCF model and the baselines.

*Table 13.* Quantum Parameters

| Parameter | Values |
|---|---|
| Natural frequency $\omega$ | Uniform$(1, 2)$ |
| Interactions coupling term $\chi$ | Uniform$(0.05, 0.1)$ |
| Cubic non-linearity coefficient $\beta$ | Uniform$(0.05, 0.2)$ |
| Damping coefficient $\gamma$ | Uniform$(0.3, 0.6)$ |
| Noise magnitude $\delta$ | Uniform$(0.01, 0.1)$ |

### E.3. Handwriting Datasets

In motion generation frameworks, learned dynamics are viewed as skills that an actuated system must not only imitate but also generalize and adapt to new scenarios. The LASA dataset (Lemme et al., 2015), a widely used benchmark, contains 2D handwritten characters but lacks self-crossing trajectories, as it was designed for first-order dynamics. To better showcase GCF's capabilities, we also include the DigiLeTs dataset (Fabi et al., 2022), which features more diverse trajectory patterns. Since the demonstrated dynamics converge to a target, we exclude HNN from the baselines, as it is limited to modeling periodic behaviors.

**Dynamics reconstruction.** Table 14 compares GCF and baselines on handwriting dynamics reconstruction. It includes four characters from the LASA dataset and four from the DigiLeTs dataset.

*Table 14.* Reconstruction error via DTWD on handwriting datasets

| Character | EF | NCDS | DHNN | GCF |
|---|---|---|---|---|
| $\sim$ | $\mathbf{0.43}_{\pm\mathbf{0.10}}$ | $0.44_{\pm 0.14}$ | $0.65_{\pm 0.15}$ | $0.44_{\pm 0.12}$ |
| $\sim$ | $0.45_{\pm 0.05}$ | $0.47_{\pm 0.10}$ | $0.53_{\pm 0.11}$ | $\mathbf{0.43}_{\pm\mathbf{0.07}}$ |
| $\cup$ | $\mathbf{0.42}_{\pm\mathbf{0.20}}$ | $0.49_{\pm 0.24}$ | $0.63_{\pm 0.50}$ | $0.44_{\pm 0.34}$ |
| $G$ | $0.85_{\pm 0.36}$ | $0.88_{\pm 0.41}$ | $0.71_{\pm 0.88}$ | $\mathbf{0.68}_{\pm\mathbf{0.24}}$ |
| $\ell$ | $2.22_{\pm 0.03}$ | $2.79_{\pm 0.09}$ | $0.82_{\pm 0.22}$ | $\mathbf{0.70}_{\pm\mathbf{0.36}}$ |
| $d$ | $3.08_{\pm 0.02}$ | $3.13_{\pm 0.07}$ | $1.08_{\pm 0.39}$ | $\mathbf{0.93}_{\pm\mathbf{0.38}}$ |
| $e$ | $2.07_{\pm 0.03}$ | $2.30_{\pm 0.05}$ | $0.78_{\pm 0.17}$ | $\mathbf{0.71}_{\pm\mathbf{0.19}}$ |
| $4$ | $2.48_{\pm 0.03}$ | $3.16_{\pm 0.10}$ | $1.10_{\pm 0.32}$ | $\mathbf{0.86}_{\pm\mathbf{0.30}}$ |

**Dynamics generalization.** Figure 16 presents generalization tests for EF, NCDS, DHNN, and GCF using two example handwritten characters. The position space is divided into a grid, with each point serving as a starting location for trajectory predictions, while the remaining state variables are initialized to zero. For DHNN, this means starting with zero velocities; for GCF, both velocities and the Lagrangian action are initialized to zero. These tests evaluate how well each model performs when dynamics are initiated outside the data manifold. The trajectories predicted by NCDS for the character LEAF_2 (Figure 16b) visibly converge and remain close to the demonstrated paths. However, NCDS operates only on first-order dynamics and fails to reconstruct second-order trajectories, as seen in Figure 16f. In contrast,

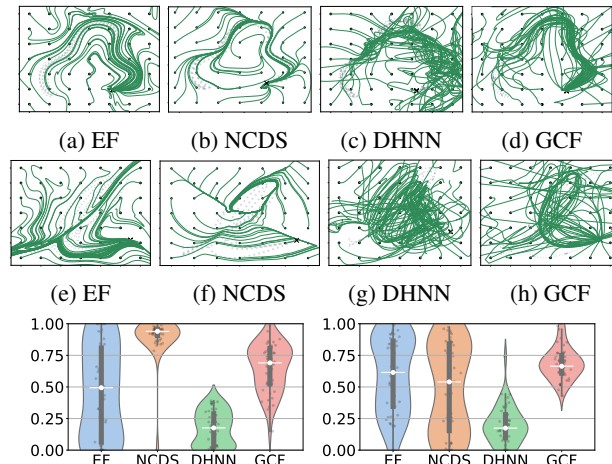

(a) EF    (b) NCDS    (c) DHNN    (d) GCF

(e) EF    (f) NCDS    (g) DHNN    (h) GCF

(i) Time spent on the data manifold for trajectories generalizing the character LEAF_2.

(j) Time spent on the data manifold for trajectories generalizing the character ELLE.

*Figure 16.* Generalization of the EF, NCDS, DHNN, and GCF methods. Predicted trajectories starting from grid points for the character LEAF_2 (16a, 16c, 16d) and for the character ELLE (16e, 16g, 16h). The gray dots (16i, 16j) represent these trajectories as points based on the ratio between time steps spent within the data manifold and the total time steps. The violin and box plots illustrate the stochastic distribution of these ratios. A high ratio indicates quick convergence. NCDS's contractive properties yield good results on LEAF_2, but struggle to converge in ELLE. In contrast, the GCF method demonstrates reliable performance on both characters, achieving the highest average ratio and lowest standard deviation, indicating consistent convergence even for trajectories initialized far from the training dataset.

GCF delivers more reliable results, maintaining consistent performance across both types of dynamics (Figures 16h, 16d). Figures 16i and 16j provide a statistical analysis of these results. Each point, representing a predicted trajectory, is computed as the ratio between time steps spent within the data manifold (in the position space) and the total time steps. As the DHNN method imposes no constraints outside the data manifold, only a small number of trajectories initiated outside the data support converge, as indicated by the large proportion of points with ratio of 0. EF guarantees asymptotic stability, ensuring that trajectories initiated outside the data support eventually converge to a target. However, it does not explicitly drive trajectories toward the data manifold, leading to a uniform distribution of ratios. NCDS improves the results for the character LEAF_2 due to its guaranteed contractive behavior. However, GCF ensures that all trajectories from both LEAF_2 and ELLE reliably converge to the data manifold. On average, these trajectories spend significantly more time within the data support than outside it, even when initiated far from the data support. Table 15 reports generalization statistics for a set of handwritten characters. GCF stands out for its ability to avoid

uncertain regions and converge to the data manifold.

*Table 15.* Generalization measured as average ratio of convergence

| Character | EF | NCDS | DHNN | GCF |
|---|---|---|---|---|
| 〰 | $0.49_{\pm 0.37}$ | $\mathbf{0.94_{\pm 0.19}}$ | $0.18_{\pm 0.13}$ | $0.69_{\pm 0.19}$ |
| 〰 | $0.59_{\pm 0.31}$ | $\mathbf{0.91_{\pm 0.22}}$ | $0.13_{\pm 0.18}$ | $0.67_{\pm 0.12}$ |
| 〰 | $0.47_{\pm 0.41}$ | $\mathbf{0.91_{\pm 0.24}}$ | $0.18_{\pm 0.10}$ | $0.62_{\pm 0.23}$ |
| G | $0.53_{\pm 0.33}$ | $\mathbf{0.93_{\pm 0.16}}$ | $0.21_{\pm 0.13}$ | $0.68_{\pm 0.14}$ |
| $\ell$ | $0.61_{\pm 0.30}$ | $0.54_{\pm 0.35}$ | $0.17_{\pm 0.15}$ | $\mathbf{0.66_{\pm 0.12}}$ |
| $d$ | $0.52_{\pm 0.40}$ | $0.50_{\pm 0.39}$ | $0.17_{\pm 0.12}$ | $\mathbf{0.64_{\pm 0.16}}$ |
| $e$ | $0.59_{\pm 0.36}$ | $0.61_{\pm 0.41}$ | $0.19_{\pm 0.13}$ | $\mathbf{0.67_{\pm 0.10}}$ |
| $4$ | $0.58_{\pm 0.34}$ | $0.51_{\pm 0.40}$ | $0.18_{\pm 0.17}$ | $\mathbf{0.65_{\pm 0.15}}$ |

**Ablation Study on Ensemble Contribution** The generalization results achieved by GCF, previously shown in Figure 16, benefit from the ensemble of contactomorphisms. Figure 17 contrasts these results from the complete GCF framework with those from a variant using a single contactomorphism. The ensemble approach, which incorporates uncertainty-aware curves, demonstrates faster convergence to the data manifold. For trajectories initiated from points already close to the data manifold, the difference in convergence rate is less pronounced, as reflected in the violin plots (Figs. 17e, 17f), where the distributions of points near 1 are similar. However, for trajectories starting farther away from the data manifold, the effect of the ensemble uncertainty is more evident, contributing to a higher overall aggregated metric. Generalization statistics for a set of handwritten characters are presented in Table 16.

*Table 16.* Generalization measured as average ratio of convergence

| Character | Single | Ensemble |
|---|---|---|
| 〰 | $0.61_{\pm 0.21}$ | $\mathbf{0.69_{\pm 0.19}}$ |
| 〰 | $0.62_{\pm 0.13}$ | $\mathbf{0.67_{\pm 0.12}}$ |
| 〰 | $0.56_{\pm 0.26}$ | $\mathbf{0.62_{\pm 0.23}}$ |
| G | $0.64_{\pm 0.15}$ | $\mathbf{0.68_{\pm 0.14}}$ |
| $\ell$ | $0.61_{\pm 0.15}$ | $\mathbf{0.66_{\pm 0.12}}$ |
| $d$ | $0.57_{\pm 0.21}$ | $\mathbf{0.64_{\pm 0.16}}$ |
| $e$ | $0.61_{\pm 0.12}$ | $\mathbf{0.67_{\pm 0.10}}$ |
| $4$ | $0.57_{\pm 0.18}$ | $\mathbf{0.65_{\pm 0.15}}$ |

**Unsafe Regions on the Handwriting Datasets** Figure 18 shows the trajectories and the distances to both the obstacle and the data manifold for the handwriting characters LEAF_2 and ELLE.

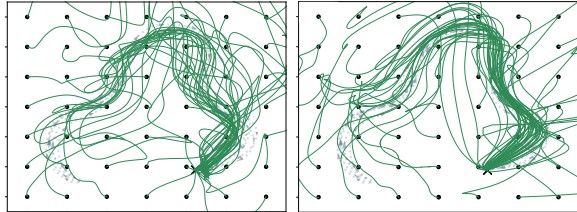

(a) Generalization with single contactomorphism (LEAF_2). (b) Generalization with the ensemble (LEAF_2).

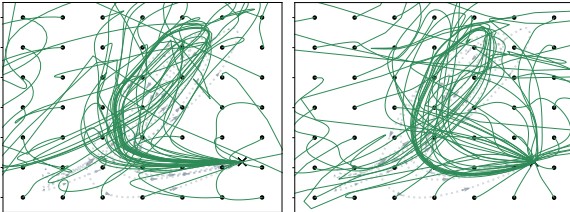

(c) Generalization with single contactomorphism (ELLE). (d) Generalization with the ensemble (ELLE).

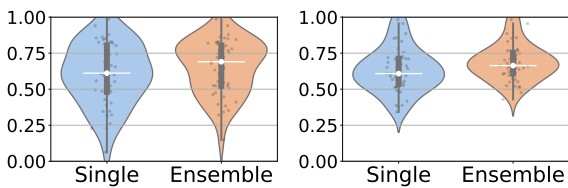

(e) Time spent on the data manifold for LEAF_2. (f) Time spent on the data manifold for ELLE.

*Figure 17.* The ablation study highlights the performance improvement in generalization capability achieved through the use of an ensemble of contactomorphisms. Plots 17a and 17b depict trajectory predictions from grid points for the character LEAF_2, while plots 17c and 17d present predictions for the character ELLE. The bottom charts (17e and 17f) summarize these trajectories by representing them as points based on the ratio of time steps spent within the data manifold to the total number of time steps. The violin plots visualize the stochastic distribution of these ratios, where a higher ratio indicates faster convergence. The use of the ensemble enhances convergence toward the data manifold, resulting in a higher ratio and a lower standard deviation.

### E.4. Robot Tests

In robotics, learning interaction tasks from demonstrations is more challenging than learning free-space motions (Scherzinger et al., 2019; Le et al., 2021). GCF excels in this context by modeling energy behaviors beyond simple motion and flexibly generalizing learned dynamics when the system encounters unforeseen regions of the state space, such as unexpected positions, velocities, or external forces. This makes GCF particularly well-suited for online robot control in interaction tasks. We apply GCF to two robotics tasks: a proof-of-concept WRAP-AND-PULL task and a more realistic DISHWASHER LOADING task.

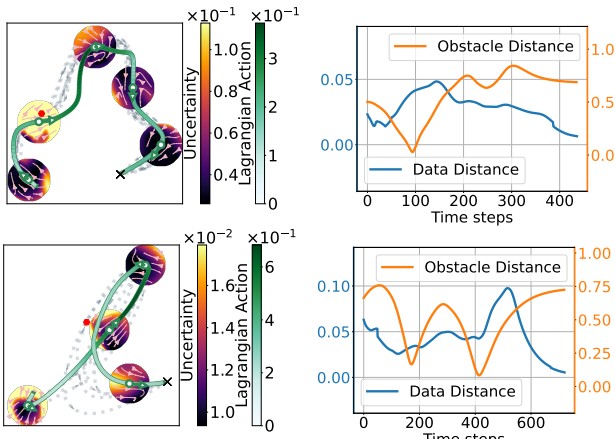

*Figure 18.* Obstacle avoidance behavior demonstrated by the LEAF_2 (top) and the ELLE (bottom) characters. Obstacles are represented as red dots.

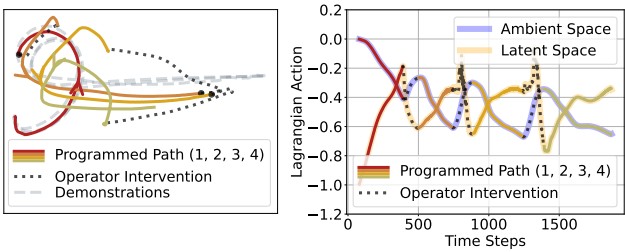

*Figure 19.* Position and Lagrangian action trajectories (*left* and *right* plots) for the unloaded WRAP-AND-PULL task when disturbed by the operator's physical intervention.

**WRAP-AND-PULL Task under Physical Perturbations**
Figure 19 shows the robot trajectories in the position and Lagrangian action spaces when physically perturbed by an operator who deviates it from its nominal path. The robot's ability to recover the wrapping and pulling behavior even under physical perturbations showcases GCF's robustness. The Lagrangian action $s$, governed by the Maupertius' principle (see Appendix D.3), decreases during the pulling task as a result of the work performed by the robot on the environment. However, it increases when the environment, represented by the operator, performs work on the robot. Consequently, the robot gains additional energy to continue progressing through the task.

**DISHWASHER LOADING Task** Household environments represent a key frontier for collaborative robotics, where robots can assist with daily tasks, improving convenience, safety, and quality of life, particularly for the elderly or individuals with limited mobility (Shafiullah et al., 2023). Figure 20 shows snapshots of the main task phases: the robot reaches for and pulls out the dishwasher basket, enabling the human operator to place a dish, then pushes the basket back in and returns to its home position. Thanks to the robustness of the GCF framework, the robot reliably converges to the

learned trajectory and successfully grasps the basket, even when disturbances (such as the operator inadvertently displacing the robot) cause deviations from the data manifold. This robustness is essential in collaborative settings where humans and robots share the same workspace and may unintentionally interfere with each other's actions. Moreover, the safe latent dynamics described in Equation (7c) allow the robot to respond appropriately when the task cannot be completed due to environmental contingencies. For instance, if an object is poorly positioned and blocks the basket from closing, the robot detects that the required energy exceeds levels encountered during training and stops exerting force, thereby avoiding potentially damaging overexertion. Also this experiment is available in the video accompanying this paper.

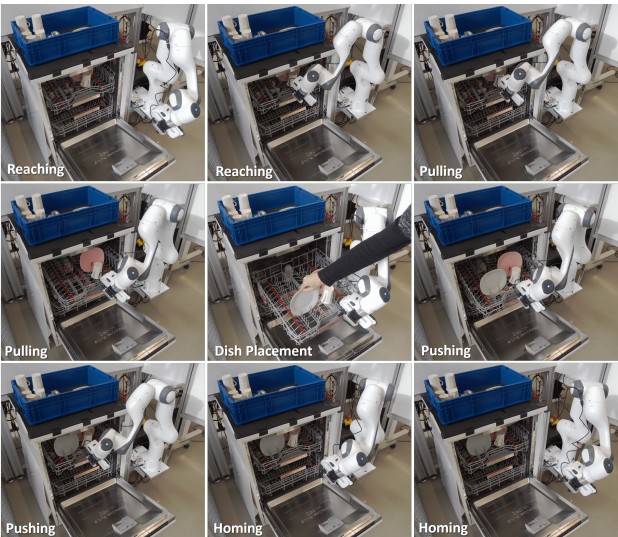

*Figure 20.* Snapshots from the collaborative DISHWASHER LOADING task, illustrating key phases: reaching the basket, pulling (opening) the basket, operator placing a dish, pushing (closing) the basket, and the robot returning to home position.

### E.5. Scalability Evaluation with Synthetic Dynamics

To evaluate how GCF scales with varying target dimensionalities and model sizes, we conduct experiments on synthetic data generated from contact Hamiltonian systems with different numbers of degrees of freedom. The dynamics are governed by a single contact Hamiltonian function parameterized by the target dimensionality $d$, ensuring consistent comparison across dimensions,

$$H = \frac{1}{2}\mathbf{p}^\top \mathbf{p} + \frac{1}{2}\mathbf{q}^\top \mathbf{q} + \sum_i^{d/2} q_{2i}q_{2i+1} + q_0 \sum_i^{d/2} q_{2i} + s. \quad (59)$$

The summation terms couple the behavior of the various degrees of freedom, producing unique dynamics for each di-

*Table 17.* Inference time (seconds) as a function of the number of contactomorphism components ($K$) and hidden units in the RFF networks ($n_f$).

| $n_f$ | $K$ | | | |
|---|---|---|---|---|
| | 5 | 10 | 15 | 20 |
| 200 | $0.194_{\pm 0.003}$ | $0.204_{\pm 0.003}$ | $0.216_{\pm 0.003}$ | $0.221_{\pm 0.002}$ |
| 350 | $0.198_{\pm 0.003}$ | $0.207_{\pm 0.003}$ | $0.218_{\pm 0.002}$ | $0.223_{\pm 0.002}$ |
| 500 | $0.195_{\pm 0.003}$ | $0.206_{\pm 0.002}$ | $0.218_{\pm 0.002}$ | $0.224_{\pm 0.001}$ |

mension. This coupling also ensures that the reconstruction task is sufficiently challenging.

We experiment with different contactomorphisms structures $\varphi_r$, by varying two key parameters: The number $K$ of individual flows $\varphi_{r_{\theta_k}}$, composing $\varphi_r$, and the number $n_h$ of hidden units, parametrizing each flow via RFF networks. The total number of parameters is computed as $K \times n_h \times n_f$, where $n_f = 3$ is the number of learning functions per individual flow. Figure 21 shows the reconstruction accuracy of

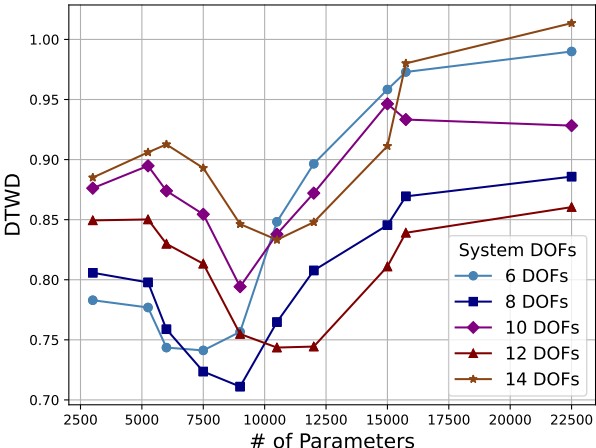

*Figure 21.* Reconstruction results achieved by the GCF method, evaluated across varying numbers of parameters used to model a contactomorphism, for systems of different dimensionalities.

different high-dimensional GCF models, as a function of the total number of parameters. Reconstruction quality, measured by DTWD, is normalized by the dimensionality value to ensure fair comparisons across systems. Note that GCF's performance remains stable as the problem dimensionality increases, with higher-dimensional target dynamics benefiting from larger architectures, at the cost of increased inference times as reported in Table 17. However, beyond a certain size, performance may decline due to overfitting, likely caused by the limited data available.

