# OpenReview forum: "Geometric Contact Flows: Contactomorphisms for Dynamics and Control"
_ICML.cc/2025/Conference — ICML 2025 poster_

### Official Review · Reviewer_89jF · 2025-03-04

**Overall Recommendation:** 3

**Summary:**

This paper introduces Geometric Contact Flows, a framework that models dynamical systems by incorporating Riemannian and contact geometry as the inductive bias. The learned latent space captures the dynamics by contactomorphically preserving the structure of the ambient space. An additional ensemble approach is proposed to model the system uncertainty. Lastly, the proposed method is evaluated on two handwritten datasets and a real-world rope-wrapping robot experiment.

**Claims And Evidence:**

The paper proposes a framework for modeling dynamical systems. The experimental results support the idea that the proposed system is able to model trajectories with intersected paths. However, it is unclear whether it is overfitting to a specific trajectory.
Secondly, the paper proposes an ensemble approach for uncertainty estimation. However, it is not experimentally verified.
Lastly, the system is claimed to be able to perform obstacle avoidance by incorporating the energy term of the obstacle, which is verified qualitatively in the experiment.

**Essential References Not Discussed:**

The experiments fall into the imitation learning domain. However, it is not compared against state-of-the-art imitation learning methods. Comparing with the state-of-the-art methods in the field can greatly strengthen the paper.

**Experimental Designs Or Analyses:**

The experimental results show the proposed GCF’s capability to learn dynamical trajectories with intersected paths. The proposed method also achieves lower reproduction errors.

However, the experimental setup of the handwritten dataset is not clearly stated. The outputs are trajectories integrated from a vector field, but what are the inputs to such a system? Is it an initial value problem with initial position and time as inputs?

Similarly, the generalization experiment is not well described either. What is the task being performed? Is the task training on two sets of trajectories and evaluating on them?

Is the network trained and evaluated on the same trajectory? If so, would it be infeasible to apply to real-world robotics tasks as it takes 4 hours to train the network? I understand that the paper is positioned as a dynamic modeling framework. In such cases, evaluating on some dynamic modeling benchmark could better demonstrate the impact of the proposed model.

**Methods And Evaluation Criteria:**

The contactomorphic idea is interesting. However, I am still not convinced why contactomorphism is a better choice compared to symplectic geometry. Additionally, the choice in (4) seems a bit arbitrary to me.

**Other Comments Or Suggestions:**

1. The paper can benefit from a flowchart that explains the overall architecture of the system. Currently, the overall flow is not explicitly clear.
2. There are some typos in the paper. Ex. Ln 17 left and ln 29 right (GFC) should be (GCF)?

**Other Strengths And Weaknesses:**

Please see the comments above.

**Questions For Authors:**

Please see the questions above.

**Relation To Broader Scientific Literature:**

The paper proposes a framework for dynamic modeling using contact geometry. It is demonstrated using several trajectory reconstruction tasks in robotics. However, the proposed method is not compared with the state-of-the-art imitation learning methods. As a result, it is unclear where the proposed method stands in the literature of imitation learning. Nevertheless, the proposed model is shown to perform better than the Euclideanizing Flows and Dissipative Hamiltonian Neural Networks.

**Theoretical Claims:**

The paper designs the latent dynamics to be contactomorphic to the ambient dynamics. It is theoretically sound but is not verified experimentally.

---

> ### Author Rebuttal · Authors · 2025-03-31
>
> > The proposed system seems able to model trajectories with intersected paths. However, it is unclear whether it is overfitting to a specific trajectory
>
> Our framework reconstructs intersecting paths in position space using the full state of the system to resolve directional ambiguities. Extensive experiments (Figs 8 and 11) confirm that GCF avoids overfitting, successfully reproducing intersecting trajectories beyond the training data support.
>
> > the paper proposes an ensemble approach for uncertainty estimation. However, it is not experimentally verified
>
> Note that all the experiments use our full framework with the ensemble of contactomorphisms. The experimental setup is detailed in Appendix C.1. Additionally, Appendix D.2 presents an ablation study on the handwriting dataset, comparing GCF with the ensemble approach against a single-contactomorphism variant. The ensemble significantly improves generalization performance (Figure 9, Table 9).
>
> > I am still not convinced why contactomorphism is a better choice compared to symplectic geometry
>
> Contact geometry naturally models both conservative and non-conservative systems, while symplectic geometry is limited to the former and require modifications to handle dissipation, making its dynamics not purely symplectic. Our approach constructs latent dynamics whose physical properties are fully encoded by contact geometry, so preserving the contact structure in the transformation to the ambient space is sufficient to propagate these properties intact.
>
> > The choice in (4) seems a bit arbitrary to me
>
> The choice of latent Hamiltonian functions (Eq. 4) is a design decision driven by the properties the user aims to preserve when generalizing in the ambient space. On the data manifold, GCF can recover the demonstrated dynamics regardless of the specific latent Hamiltonian function, while outside this manifold, the latent dynamics structure acts as a physical bias to guide generalization.
>
> > The paper designs the latent dynamics to be contactomorphic to the ambient dynamics. It is theoretically sound but is not verified experimentally
>
> We can verify this by evaluating the contact transformation equations that map the ambient state $(q, p, s)$ to the latent state $(\hat{q}, \hat{p}, \hat{s})$, as established in [Bravetti et al.](https://arxiv.org/abs/1604.08266):
>
> $$
> p_i \frac{d \hat{s}}{d s} - p_i \hat{p}_i \frac{d \hat{q}}{d s_i} = - \frac{d \hat{s}}{d q_i} + \hat{p}_i \frac{d \hat{q}}{d q_i};
> $$
>
> $$
> \frac{d \hat{s}}{d p_i} - \frac{d \hat{q}_i}{d p_i} = 0.
> $$
>
> These partial derivatives are elements of the contactomorphism Jacobian. By evaluating these conditions, we consistently observe an error lower than $1 \cdot 10^{-5}$, confirming that the transformation is contactomorphic.
>
> > the experimental setup of the handwritten dataset is not clearly stated
>
> In the handwriting experiment, the input to our framework is the current system state, while the output is the next state. This prediction is repeated at each time step to reconstruct the full dynamics. Our approach treats the dynamical system as autonomous, excluding time as an explicit input.
>
> > the generalization experiment is not well described
>
> In the generalization experiments, we use models trained to reconstruct a specific dynamical trajectory but initialize the predictions from states far outside the training data distribution. This evaluates model performance on states unseen during training. Specifically, as shown in Fig. 8, we initialize predictions from a grid of points in the position space, while setting the remaining state variables to zero.
>
> > Is the network trained and evaluated on the same trajectory? If so, would it be infeasible to apply to real-world robotics tasks as it takes 4 hours to train the network?
>
> Yes, the network is trained and evaluated on the same dynamics. However, reproduction goes beyond imitation, allowing adjustments like obstacle avoidance, and ensuring convergence to the learned dynamics under different initial conditions or external disturbances. In robotics, many tasks are inherently repetitive, allowing a dynamical primitive learned in four hours to be reliably reused across various scenarios, with the robustness of the framework ensuring reliable adaptation and performance.
>
> > evaluating on some dynamic modeling benchmark could better demonstrate the impact of the proposed model
>
> We incorporated new evaluations on material deformation and quantum dynamics simulation, detailed in our reply to Reviewer F9hb.
>
> > The proposed method is not compared with the state-of-the-art imitation learning methods
>
> We kindly refer the reviewer to our response to Reviewer sRBK, where we emphasize the rationale behind our baselines choice. Additionally, in response to the reviewer's request, we included two new baselines: NCDS and HNN, the latter in the new experiments.
>
> > Paper can benefit from a flowchart
>
> https://drive.google.com/file/d/1xCe6Rh7FU0ZEwVdKfEw16cNSa1MKkS-R/view?usp=sharing

---

### Official Review · Reviewer_sRBK · 2025-03-10

**Overall Recommendation:** 3

**Summary:**

This paper introduces a geometric contact flows model based on Riemannian and contact geometry, which introduces a robust and interpretable inductive bias over the previous MLP based methods. Furthermore, the authors propose a novel framework to learn latent dynamics of contactomorphisms and generalization mechanism based on Riemannian geodesics, which also improves the model robustness. Experiments show superior performance over baseline methods on multiple tasks such as reconstructing handwriting dynamics and robotic interactions.

**Claims And Evidence:**

Yes

**Essential References Not Discussed:**

No

**Experimental Designs Or Analyses:**

Yes

**Methods And Evaluation Criteria:**

Yes, methods are evaluated on LASA and DigiLeT datasets for handwriting trajectory reconstruction. Experiments detailed are included in supplementary materials.

**Other Comments Or Suggestions:**

NA

**Other Strengths And Weaknesses:**

1. The author claims that introducing Riemannian geometry as inductive bias shows improved robustness over simple MLP with no additional priors. However, I wonder how is the baseline models perform if the model is equipped with a stronger priors, particularly those captured from differentiable simulators, see some works below [R1 - R3]. While I understand the contact dynamics in these works may not be directly relatable, it would be interesting to see a results of a similar baseline or discussion on these methods.

2. In quantitative experiments the authors compared with EF and DHNN without providing a detailed introduction of these two methods. Why are these two baselines chosen and are there other more recent works comparable?

3. While the section 3 - 4 introduces most concepts from contact geometry, I find limited designs on the learning framework, in particular how is the network designed and the geometry priors fused? I also do not find enough experiments ablating the development of the modules inside the neural networks, make it hard to evaluate its contribution in the ML side.

[R1] SimPoE: Simulated Character Control for 3D Human Pose Estimation
[R2] Residual Force Control for Agile Human Behavior Imitation and Extended Motion Synthesis
[R3] DeepSimHO: Stable Pose Estimation for Hand-Object Interaction via Physics Simulation

**Questions For Authors:**

Please see above weakness

**Relation To Broader Scientific Literature:**

The application of the system can generally applied to contact related tasks such as robots-object interaction and trajectory synthesis. No further broader impacts identified.

**Theoretical Claims:**

The paper is not a theoretical paper.

---

> ### Author Rebuttal · Authors · 2025-03-31
>
> > The application of the system can generally applied to contact related tasks such as robots-object interaction and trajectory synthesis. No further broader impacts identified.
>
> We clarify that the contact Hamiltonian biases in our framework extend beyond interaction tasks in control-based approaches. They represent fundamental physical principles applicable to modeling dissipative mechanical systems, thermodynamic processes, and quantum dynamics. To highlight this broader impact, we introduce two additional physical reconstruction experiments (spring mesh and quantum system), as detailed in our response to Reviewer F9hb.
>
> > I wonder how is the baseline models perform if the model is equipped with a stronger priors, particularly those captured from differentiable simulators.
>
> The comparison with a simple MLP in our methodology serves to motivate our approach rather than act as the main baseline. As detailed below, the baselines used in our results are equipped with strong priors.
>
> The referenced differentiable simulators learn policies for physically meaningful behaviors by leveraging physical simulation during training to evaluate policy performance. Since they introduce physical biases during training, there is no guarantee that the learned policies will remain physically consistent when generalized to new scenarios. In contrast, our approach (and the baselines we consider) embeds biases directly within the network structure itself.
>
> > Why are these two baselines chosen and are there other more recent works comparable?
>
> The selected baselines are well-known for incorporating biases in learning dynamical systems, introducing features that our approach successfully recovers and generalizes:
> - Encoding desirable properties (e.g., periodicity or target convergence) in the dynamics through diffeomorphisms (EF).
> - Embedding physical relationships between the components of the system's state through Hamiltonian dynamics (DHNN).
>
> Our approach extends EF's idea of transforming latent dynamics using diffeomorphisms by considering second-order dynamics and by introducing a (more general) contact Hamiltonian structure in the diffeomorphisms to preserve conjugate pair relationships. DHNN achieves second-order modeling by embedding pure Hamiltonian dynamics in the network structure, but it lacks the ability to enforce desirable properties in the learned dynamics. The comparison with these baselines in our experiments highlights the importance of both biases. A more recent work (2024) aligned with our philosophy is [NCDS](https://openreview.net/forum?id=Q5N3P0SMRr), which extends the properties of a learned latent contractive system to the ambient space using structure-preserving transformations. To strengthen our baseline comparison, we included this approach in the [experiments](https://drive.google.com/file/d/1RcE8tb_gQxbQ3e0ERz-BUPIzk9Lk9uq-/view?usp=sharing).
>
> > how is the network designed and the geometry priors fused?
>
> The network $\varphi_r$ (Eq. 6), is implemented as a sequence of chained transformations $\varphi_{r_k}$ (Eq. 7). Each of these transformations consists of three steps (Eq. 24), which updates the initial state by integrating the vector field associated to the contact Hamiltonian $H_{r_k}$ (Eq. 8). This Hamiltonian is composed of three learning functions $M(p), V(q), F(q)$, parametrized by RFFNs. Therefore, the network $\varphi_r$, which integrates the dynamics of a sequence of Hamiltonians, characterizes a contact flow.
>
> > I also do not find enough experiments ablating the development of the modules inside the neural networks
>
> We address the reviewer's suggestion by introducing three additional ablations:
>
> - Is the contact structure truly necessary? We assess its importance by examining the issues that arise when replacing contactomorphisms with naive diffeomorphisms, implemented similarly to EF. The disruption of physical coherence manifests in poor reconstruction and generalization performance: [contact-structure-ablation](https://drive.google.com/file/d/1OV8Wonn9_ITfwHKmRxJMrbULw0geT2Cq/view?usp=sharing)
>
> - How does the Hamiltonian function (Eq. 8) or its parametrization affect GCF performance? We test variations of the Hamiltonian function and compare different architectures for parameterizing the learning functions. The Hamiltonian that incorporates all functions achieves the best reconstruction results, while RFFN proves to be the best parametrization choice: [learning-functions-ablation](https://drive.google.com/file/d/1RiYCiGLknc4DHntgnqejI5Vx3ZJr9zRz/view?usp=sharing)
>
> - Why does the loss function have this specific form? We examine the effect of the second loss term (Eq. 9) by varying its scaling factor and analyzing the resulting performance differences. The study finds an ideal range where improved coherence of the latent space enhances reconstruction in the ambient space: [loss-term-ablation](https://drive.google.com/file/d/1hDVZR168YKjaUdgO12ALfZ-NbIfkconA/view?usp=sharing)

---

### Official Review · Reviewer_F9hb · 2025-03-10

**Overall Recommendation:** 3

**Summary:**

The paper proposes to learn in the latent contact Hamiltonian space to inject inductive biases and encoding desirable physical properties. Additionally, the paper developed an ensemble method that aims to identify the unseen states and drive the dynamics to avoid these states. Experiments in character writing and robot-object manipulation verified the proposed method outperforms two previous works.

## update after rebuttal
I appreciate the authors' additional experiments in the rebuttal. The new simulation and robot experiments demonstrate that the proposed method can learn dynamics with more variations (the spring-mesh experiments) and address real-world problems (the robot-dishwasher experiments). I raised the score accordingly.

**Claims And Evidence:**

The claims in the paper are supported by experiments.

**Essential References Not Discussed:**

No

**Experimental Designs Or Analyses:**

These two experiments in the paper are too simple. These experiments are state based, and have no variations. Consider image-based, or language-based, or the tasks involving agent-env interaction (Artari game or DeepMind control suite, https://github.com/openai/gym).

**Methods And Evaluation Criteria:**

The evaluation tasks (Handwriting Datasets, Robotic Task) seem like simple trajectory generalization tasks. These tasks are state based and have very limited variations (4 characters and 1 robot trajectory). Testing on more complex datasets could be more convincing (e.g., image generation, more complex robotic manipulation tasks).

**Other Comments Or Suggestions:**

The acronyms of the method should be GCF, but there are multiple places typed as GFC.

**Other Strengths And Weaknesses:**

Strength:
Theoretically valid: adding inductive bias and learning physically preserving properties are meaningful for dynamics learning.
Weakness:
The experiments in the paper are too simple.

**Questions For Authors:**

Does the proposed method scale to more complex, real-world problems, for example, learning rigid body interactions, or learning dynamics of one / two-link pendulum?
Does the proposed method scale to high-dimensional states, for example, images (videos) or point-clouds' dynamics?
Could the author compare the proposed method with more recent trajectory generation methods, for example, diffusion models[1] [2]?

[1] Cheng Chi, et al, Diffusion Policy: Visuomotor Policy Learning via Action Diffusion.
[2] Michael Janner, et al, Planning with Diffusion for Flexible Behavior Synthesis.

**Relation To Broader Scientific Literature:**

Learning dynamics is important in the learning world models, and could benefit the learning community in general. However, the proposed method demonstrated limited dynamic learning ability in two simple tasks.

**Theoretical Claims:**

No proofs in the paper. In terms of the idea that ensembling contactomorphisms, it does not only reflect the data support, it could also indicate the randomness of the data. In some scenarios randomness may be preferred, e.g., asking a household robot to mix food ingredients for cooking.

---

> ### Author Rebuttal · Authors · 2025-03-31
>
> > The evaluation tasks seem like simple trajectory generalization tasks. These tasks are state based and have very limited variations (4 characters and 1 robot trajectory).
>
> > Does the proposed method scale to more complex, real-world problems, for example ... ?
>
> Yes, as emphasized in the introduction, the proposed method aims at modelling complex non-conservative dynamical systems. Its applications extend beyond trajectory synthesis for control-related tasks (robotics) to a broader modeling of intricate physical phenomena. The reviewer’s suggestions fall within the natural scope of our framework, and we thus expand our evaluation with other dynamic modeling benchmarks to further demonstrate its capabilities.
>
> To clarify, our handwriting dataset experiments were conducted on eight characters, not four as pointed out by the reviewer. The results reported in Tables 1 and 2 in the paper represent a subset of our findings, while the complete set is detailed in Tables 6 and 7.
>
> Added experiments:
>
> - We consider a 60-dimensional [dataset](https://github.com/karlotness/nn-benchmark) describing the dynamics of a 2D square grid of nodes connected by springs. Predicting the dynamics of mesh nodes closely parallels finite element modeling of material deformation. The coupling of multiple springs leads to complex large-scale deformations and oscillations. **GCF reduces reconstruction error by 57% across 20 different dynamic simulations with varying initial conditions**. Experiment parameters and results are available here: [sping-mesh-experiment](https://drive.google.com/file/d/1w4zLbyYc8cx0TV1hpJjqlTPG1YQnxcLW/view?usp=sharing).
>
> - We also use GCF to reconstruct the expected dynamics of a single-mode bosonic system, simulated with a stochastic Schrödinger equation. We generated 20 trajectories by integrating the equation over 8 seconds. **GCF reduces reconstruction error by 60% in modeling the system’s evolution**. Experiment parameters and results are available here: [quantum-experiment](https://drive.google.com/file/d/16ly3ixFi01tDxxZW-jkPfu2_ZBXIfIvM/view?usp=sharing).
>
> > Does the proposed method scale to high-dimensional states, for example, images (videos) or point-clouds' dynamics?
>
> As demonstrated in the high-dimensional tests in Appendix D.5 and our new spring-mesh experiment, our method scales effectively to high-dimensional states.
> Besides, GCF has the potential to be integrated with state estimation methods that extract system variables (e.g., position and velocity) from high-dimensional representations such as images, videos, and point clouds.
>
> However, as this paper introduces an entirely new methodology, we focused on establishing its core capabilities and did not explore such integrations within this work.
>
> > Could the author compare the proposed method with more recent trajectory generation methods, for example, diffusion models?
>
> As clarified above, our approach models complex non-conservative dynamical systems with physics-informed biases, capturing second-order dynamics, including self-intersections. In contrast, the referenced diffusion model-based methods are path planners, generating motion policies based on first-order Langevin dynamics, which by design cannot capture second-order systems. While self-intersecting trajectories could emerge in diffusion-based approaches due to their stochastic nature, these intersections result from the multimodal distribution of sampled actions, not from the correct modeling of physical laws. Moreover, unlike GCFs, the provided references build on diffusion models without physics-informed bias, preventing them from guaranteeing specific dynamic behaviors.
>
> > Testing on more complex datasets could be more convincing (e.g., image generation, more complex robotic manipulation tasks)
>
> Following the reviewer's suggestion, we tested GCF in a dishwasher-loading task. The robot handles disturbances while pulling out the basket and closing it. If obstructed, it detects excessive force and stops. Snapshots are available here: [dishwasher-experiment](https://drive.google.com/file/d/1o-jpCiV6rE21_5mSLWgK5gWgEOZd3P26/view?usp=sharing), [dishwasher-variants](https://drive.google.com/file/d/1t_bDq2wLEL8TMrh3no6jPOzGAGgax1Hh/view?usp=sharing).
>
> > No proofs in the paper. In terms of the idea that ensembling contactomorphisms, it does not only reflect the data support, it could also indicate the randomness of the data.
>
> In our framework, comparing dynamics predictions from different contactomorphisms helps determine whether GCF can infer a principled dynamics from sufficient training data or should prioritize convergence to the data manifold when information is lacking. Regarding the reviewer's mention of "randomness of data," it is unclear whether this refers to noise, multimodality, or another aspect. The cooking example does not clarify further. We kindly ask the reviewer to confirm that our interpretation of their feedback is correct, or to elaborate it otherwise.

---

### Decision · Program_Chairs · 2025-05-01

**Decision:**

Accept (poster)

**Comment:**

It is an interesting and innovative idea, which is also reflected in the reviewers' consensus (cf., specific reviews below)